# Multimodal gradients unify local and global cortical organization

Yezhou Wang ®[1] ✉, Nicole Eichert ®[2], Casey Paquola[3],
Raul Rodriguez-Cruces ®[1], Jordan DeKraker[1], Jessica Royer ®[1],
Donna Gift Cabalo[1], Hans Auer ®[1], Alexander Ngo[1], Ilana R. Leppert[1],
Christine L. Tardif[1,4], David A. Rudko[1,4,5], Robert Leech ®[6], Katrin Amunts ®[7,8],
Sofie L. Valk ®[3,9], Jonathan Smallwood ®[10], Alan C. Evans ®[1] &
Boris C. Bernhardt ®[1] ✉

Functional specialization of brain areas and subregions, as well as their integration into large-scale networks, are key principles in neuroscience. Consolidating both local and global perspectives on cortical organization, however, remains challenging. Here, we present an approach to integrate inter- and intra-areal similarities of microstructure, structural connectivity, and functional interactions. Using high-field in-vivo 7 tesla (7 T) Magnetic Resonance Imaging (MRI) data and a probabilistic *post-mortem* atlas of cortical cytoarchitecture, we derive multimodal gradients that capture cortex-wide organization. Inter-areal similarities follow a canonical sensory-fugal gradient, linking cortical integration with functional diversity across tasks. However, intra-areal heterogeneity does not follow this pattern, with greater variability in association cortices. Findings are replicated in an independent 7 T dataset and a 100-subject 3 tesla (3 T) cohort. These results highlight a robust coupling between local arealization and global cortical motifs, advancing our understanding of how specialization and integration shape human brain function.

Understanding how the spatial organization of the human brain gives rise to cognitive functions is a challenging, yet fundamental goal for human neuroscience[1]. Complex brain networks at multiple scales arise from overlapping variations in cortical microstructure, function, and connectivity[2,3]. This network involves both the global integration and local specialization of cortical regions, giving rise to distributed functional communities that enable complex computations[4–7]. Global integration prominently manifests within higher-order systems, notably the transmodal association cortex, which engages in increasingly abstract and self-generated cognition[8–14]. In contrast, local functional specialization is more frequent in sensory and motor regions that interact more closely with the here and now[6,8,15,16]. The interplay between contrasting local and global motifs contributes to the hierarchical organization of the brain, underpinning segregated and integrative information processing across different cognitive functions.

Mapping structural and functional descriptors to define discrete brain areas is essential for understanding hierarchical brain organization at macroscale. Constructing precise maps of cortical areas has

[1]McConnell Brain Imaging Centre, Montreal Neurological Institute and Hospital, McGill University, Montreal, QC, Canada. [2]Wellcome Centre for Integrative Neuroimaging, Centre for Functional MRI of the Brain (FMRIB), John Radcliffe Hospital, University of Oxford, Oxford, UK. [3]Institute of Neuroscience and Medicine (INM-7), Forschungszentrum Jülich, Jülich, Germany. [4]Department of Biomedical Engineering, McGill University, Montreal, QC, Canada. [5]Department of Neurology and Neurosurgery, McGill University, Montreal, QC, Canada. [6]Department of Neuroimaging, Institute of Psychiatry, Psychology and Neuroscience, King's College London, London, UK. [7]Institute of Neuroscience and Medicine (INM-1), Forschungszentrum Jülich, Jülich, Germany. [8]C. and O. Vogt Institute of Brain Research, Medical Faculty, University Hospital Düsseldorf, Heinrich Heine University of Düsseldorf, Düsseldorf, Germany. [9]Cognitive Neurogenetics, Max Planck Institute for Human Cognitive and Brain Sciences, Leipzig, Germany. [10]Queen's University, Kingston, ON, Canada. ✉e-mail: yezhou.wang@mail.mcgill.ca; boris.bernhardt@mcgill.ca

been a long-standing objective in neuroanatomy, as it reduces complexity and bias when studying brain regions and inter-regional relationships[3,17–19]. Cytoarchitecture, encompassing the arrangement, distribution, composition, and layering of cells, has emerged as a gold standard to define areas[20]. Microstructural insight from this *post-mortem* approach enhances our understanding of connectivity patterns and can illuminate the role of a region in cortical functioning. However, the microstructural patterns in the human brain and their relation to cortical function remain challenging to address in a systematic manner due to the constraints of invasive techniques. Recently, a 3D probabilistic atlas of human brain cytoarchitecture[19] has been made available, offering valuable opportunities for examining both the micro- and macro-organization of the human brain, and for contrasting local specialization with global integration. It can, thus, help to guide investigations of structure-function association across the cortical manifold within defined areal subunits.

Gradual changes in cortical organization at macroscale have been described as well, even in early work[21]. While atlases of cortical areas discretize the brain into non-overlapping constituents, recent advances emphasize a potential complementary utility of using dimensional descriptions of the cortex. Such a perspective can help to account for cytoarchitectonic changes within an area (e.g., ocular dominance columns, border tuft and fringe area in the visual cortex), as well as changes occurring at macroscale between different cortical areas[20]. Recent work in computational anatomy has confirmed that such complementary descriptions of macroscale cortical organization can be derived from eigenvector decomposition of cortex-wide similarities in neural patterns (commonly known as *cortical gradients*[22–24]). These gradients differentiate cortical systems in an ordered and continuous manner, and can be applied to different types of neural data, both based on in vivo neuroimaging as well as *post-mortem* histology. Notably, converging hierarchical trends, spanning from sensory to transmodal regions, were observed across microstructural[7,23,25–27] and functional gradients[22,28]. These multiple dimensions can effectively capture nuanced patterns of cortical organization[26], and may provide synergy in understanding subregional heterogeneity and functional multiplicity of different cortical areas[29]. More broadly, gradient mapping techniques have robustly differentiated transmodal association cortices from primary sensory/motor systems, mirroring their hierarchical contributions to cognition. In effect, such gradients have been found to align with functionally relevant properties, including disproportionate expansion during primate evolution[30–32], reduced heritability and increased experience-dependent plasticity[27,33], increased network idiosyncrasy[34], and the balance of internal vs. externally oriented processing[35–37]. Moreover, gradients may help to potentially account for recent findings showing that functional activation patterns, as well as functional connectivity, can shift across different contexts and individuals[38–40].

Local vs. global organization can be interrogated at the level of microstructure (e.g., cytoarchitecture), connectivity, and function. In this context, MRI serves as an ideal technique to bridge structure and function across varying spatial scales[41,42]. T1 relaxometry is sensitive to cortical microstructure and myelination[43,44], diffusion MRI tractography approximates structural connectivity[45,46], and resting-state functional MRI (rs-fMRI) has delineated macroscopic functional networks[4,47,48]. Notably, while conventional MRI acquisitions at field strengths of 3 T and below may have limitations in terms of resolution and signal, recent in vivo studies that have moved to high fields of 7 T and above have benefitted from enhanced resolution, sensitivity, and biological specificity[49–51]. In addition, imaging paradigms that combine multiple MRI datasets acquired across different scanning sessions in a given individual have been shown to further increase precision for the analysis of microstructure[52], connectivity[53,54], and function[53,55]. Several of such "precision neuroimaging" datasets have already led to an advanced characterization of functional systems[56,57] or fostered

enhanced microstructural modeling, but previous precision imaging datasets were either prioritizing functional or structural imaging acquisitions, and rarely both in the same subjects. Moreover, prior precision imaging investigations were mainly carried out at 3 T. In this study, we expand this work by leveraging a recently introduced precision neuroimaging (PNI) dataset[58], which combines repeated high-resolution structural and functional acquisitions at 7 T, offering an opportunity to interrogate cortical organization in the living human brain with high sensitivity and specificity.

The current work examined the interplay of local cortical arealization and global integration. Leveraging probabilistic cytoarchitectonic maps of the recently disseminated Julich-Brain atlas[19], we subdivided the cortex into 228 areas. In those, we profiled microstructural, structural, and functional gradients derived from repeated 7 T MRI scans. We then examined how multimodal gradient profiles differed across areas. As local-global cortical organization is presumably tied to cognitive functional architecture, we cross-referenced our maps to multiple fMRI tasks conducted in the same participants, and in particular studied the relation between inter-areal gradient profiles and functional diversity across different tasks acquired in the same subjects. The main analyzes were replicated using different parcellation atlases and datasets to validate the robustness and reliability of our findings. By integrating measures of cortical cytoarchitecture with multimodal high-definition MRI, our work sheds light on local-global cortical organization and advances our understanding of cortical structure-function relationships.

## Results
### Bridging local and global cortical organization
We constructed high-resolution cortex-wide connectomes, encompassing microstructural profile covariance (MPC)[23], structural connectivity (SC)[59], and functional connectivity (FC)[22,60], in 10 healthy adults who underwent three repeated multimodal MRI scans at 7 T (Fig. 1A). We estimated connectome eigenvectors that characterized spatial gradients of MPC, SC, and FC, focusing on the first five gradients in each modality, which explained most of the variance (MPC: 31%; SC: 18%; FC: 25%). In line with prior work[23], the principal MPC gradient was anchored on one end by primary sensory areas and on the other end by paralimbic regions. The principal SC gradient exhibited anterior-posterior axis, clearly dividing the cortex into two parts bounded by sensorimotor areas, as reported previously[61]. The first FC gradient differentiated sensory and motor cortices from the default mode network, recapitulating earlier work[22,60]. Other gradients were also in keeping with prior reports (Fig. 1A)[23,59–61]. To ensure equal contribution from each modality, we normalized the gradients within each modality and then averaged them in areas derived from the Julich-Brain atlas (Fig. 1B). The Julich-Brain atlas enables the investigation of structure-function association within cytoarchitectonically defined cortical subunits, though it does not cover the full cortex (please see the replication analyzes, where a cortex-wise parcellation was used). This process generated an area-wise gradient profile matrix. This matrix captures most of the information in multimodal connectomes, making it an ideal measure for investigating inter-areal differences and similarities.

### Inter-areal patterns of local-global integration
We examined the similarity and differences of gradient profiles between different areas. To this end, we first conducted a principal component analysis (PCA) on the multimodal gradient profiles. These gradient profiles were reordered based on their principal component (which explained 29.6% of the variation), following a sensory-fugal axis anchored on prefrontal/cingulate regions on the one end and central/occipital regions on the other (Fig. 1C). This approach integrates salient features of its constituents (i.e., the individual MPC, SC, and FC gradients) in a synoptic manner, suggesting an overarching principle

of cortical organization across multiple modalities. The reordered gradient profiles located in the middle and at the two ends of the main axis were examined and showed diverse patterns (Fig. 1C). Specifically, we observed that the bottom region of the PCA axis, corresponding to sensorimotor areas, exhibited lower gradient z-scores for MPCG1 and FCG1 and higher scores for SCG1 and SCG2. This pattern suggests that the sensorimotor network represents one end of the hierarchy across all modalities. At the opposite end of the PCA axis, regions in the inferior frontal sulcus showed the reverse pattern, representing the other extreme of the hierarchy. In contrast, the middle region displayed a relatively uniform z-score distribution, suggesting its role in linking higher-order and lower-order regions for multimodal information processing.

To further quantify area-to-area differences, we computed an inter-areal cosine distance matrix based on the original multimodal gradient profiles (Fig. 2A). The mean value of each row in this matrix indicates the overall dissimilarity of a given area from all other areas in terms of the multimodal gradient profiles. To identify cortical areas with significantly higher/lower dissimilarity compared to all other areas, we conducted spatial permutation tests (1000 permutations) that randomly rotated the Julich-Brain atlas on a sphere[62]. We found significant and highest inter-areal dissimilarity in sensorimotor regions ($p_{spin} < 0.05$, false discovery rate (FDR) correction; Fig. 2A), indicating that these areas are the most unique across the cortex in terms of their multidimensional gradient profiles. Conversely, we observed the lowest dissimilarity in insular and fronto-temporal regions after spatial permutation tests ($p_{spin} < 0.05$, FDR correction). This suggests that sensorimotor areas are most segregated within the overall cortical hierarchy, supporting functional specialization. Although inter-areal dissimilarity was found to correlate with temporal signal-to-noise ratio

## A. Generation of multimodal cortical gradients

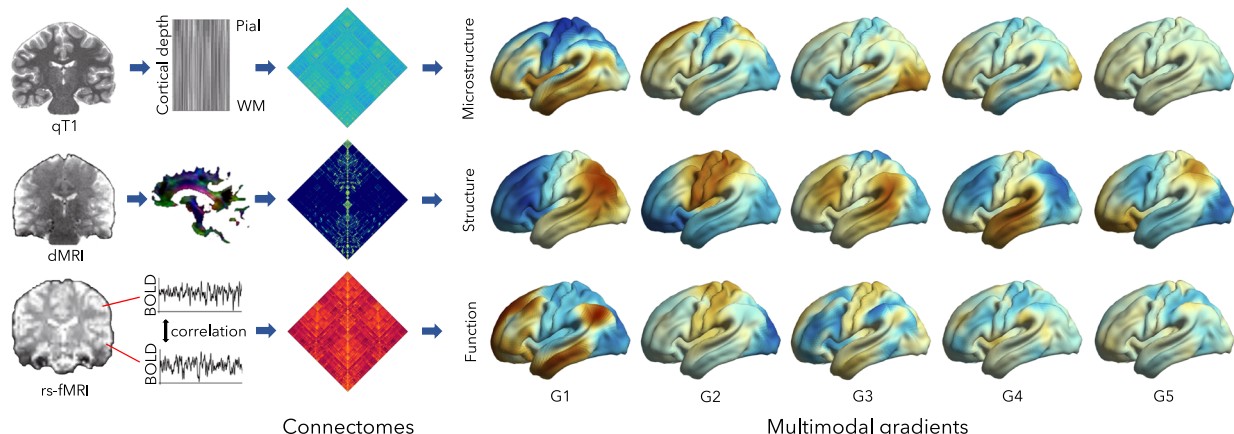

## B. Julich-Brain Atlas   C. Area-wise gradient profiling

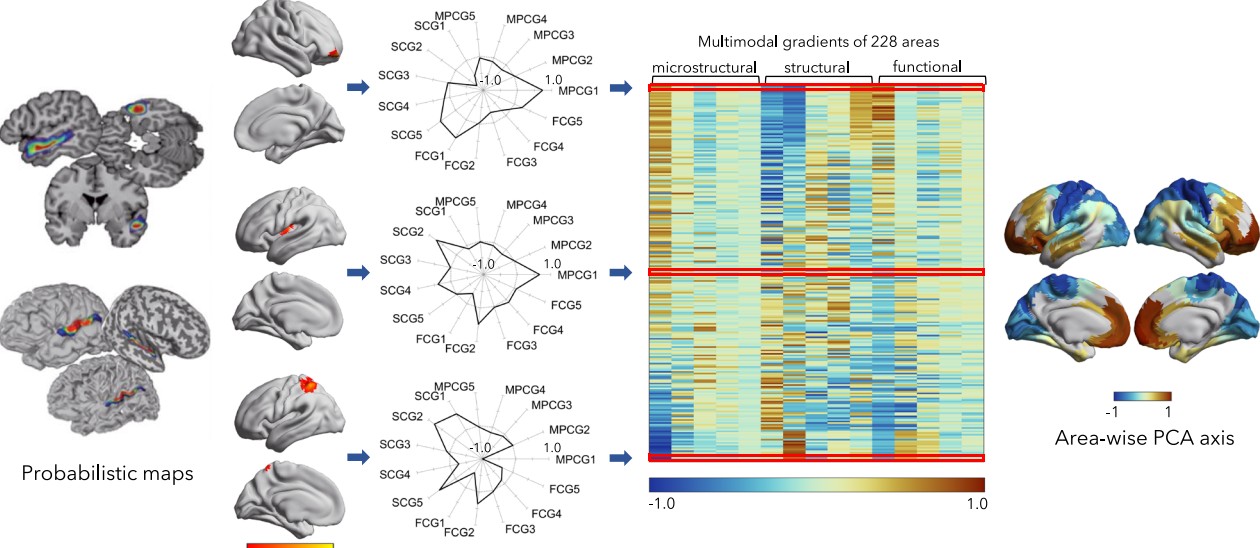

**Fig. 1 | Integration of global cortical gradients with cortical arealization.**
**A** Cortex-wide connectomes were constructed from microstructural profile covariance (MPC)[23], structural connectivity (SC)[59,61], and functional connectivity (FC)[22,60] at a vertex-level based on repeated 7 T MRI. We applied non-linear dimensionality reduction techniques[129] to each connectome and aggregated the first five eigenvectors/gradients. **B** Probabilistic area definitions were derived from the Julich-Brain atlas[19], a *post-mortem* cytoarchitectonic atlas based on the mapping of areas of ten postmortem brains, and their superimposition in MNI space. Please note that this probabilistic atlas does not cover the entire cortex. **C** We averaged vertex-wise gradients in each of the 228 areas, producing area-specific multimodal gradient profiles. These gradient profiles were reordered according to their principal component to assess inter-areal similarity. Left panel: the reordered gradient profiles located in the middle and at the two ends of the main axis were visualized in spider plots. Source data are provided as a Source Data file. Middle panel: the reordered multimodal gradient profiles. Right panel: the first principal component from the original multimodal gradient profiles. Abbreviation: PCA principal component analysis, WM white matter.

### A. Inter-areal dissimilarity patterns

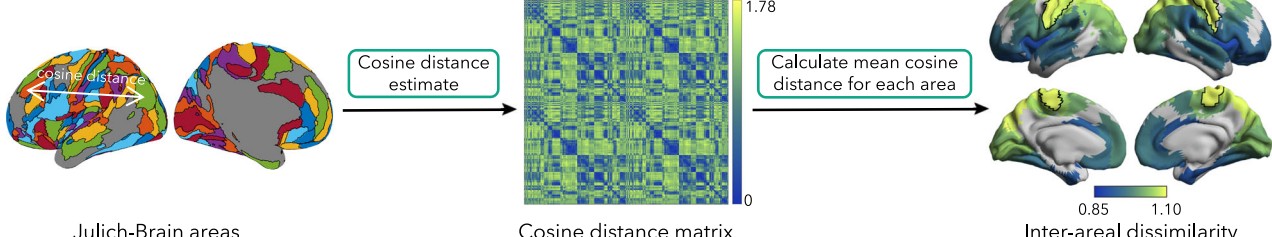

### B. Inter-areal dissimilarity in cortical hierarchies and associations with histological gradient

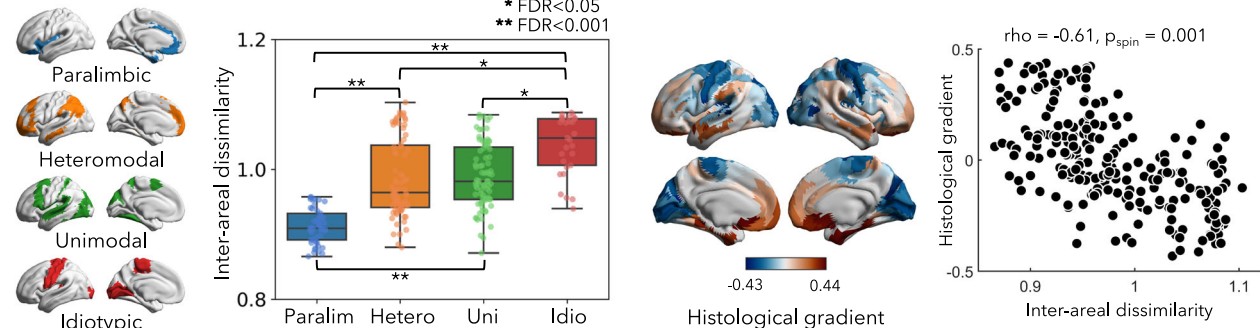

### C. Intra-areal dissimilarity patterns

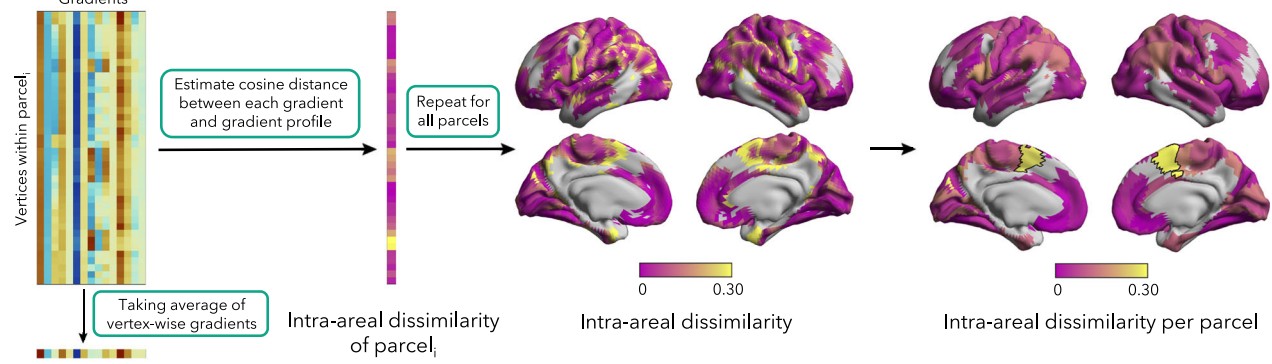

### D. Intra-areal dissimilarity in cortical hierarchies and associations with histological gradient

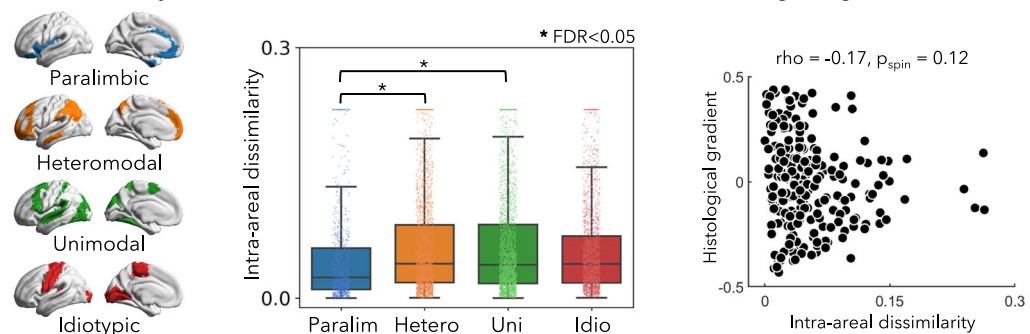

(tSNR) map ($r = 0.66$, $p_{spin} < 0.001$; Supplementary Table 1), no significant correlations were observed with other technical imaging metrics, including the B1+ field map ($r = −0.05$, $p_{spin} = 0.22$), the contrast-to-noise ratio (CNR) map from T1 scans ($r = 0.12$, $p_{spin} = 0.28$), or the SNR map from diffusion scans ($r = −0.11$, $p_{spin} = 0.34$). To investigate functional integration patterns, we estimated the participation coefficient (PC)[63] and global FC strength. Positive, but non-significant, correlations were observed between these metrics and inter-areal dissimilarity (PC: rho = 0.13, $p_{spin} > 0.05$; global FC strength:

rho = 0.32, $p_{spin} > 0.05$). Similar findings were observed when calculating these metrics based on group-level resting-state FC (PC: rho = 0.13, $p_{spin} > 0.05$; global FC strength: rho = 0.36, $p_{spin} > 0.05$).

To gain a deeper understanding of these patterns, we investigated the distribution of inter-areal dissimilarity across four cortical hierarchical levels derived from a prior taxonomy of the primate brain proposed by Mesulam[64]. Using a two-sample *t*-test, we compared overall inter-areal dissimilarity between each pair of cortical hierarchical levels (i.e., paralimbic, heteromodal, unimodal, and idiotypic).

**Fig. 2 | Inter- and intra-areal heterogeneity. A** We calculated the cosine distance between each pair of cortical areas and computed the mean value for each area. To identify cortical areas with the highest and lowest inter-areal dissimilarity, we conducted 1000 permutation tests[62]. Regions with significantly higher inter-areal dissimilarity compared to other areas after applying FDR correction were highlighted using black boundaries ($p < 0.05$). **B** The *left panel* illustrates the distribution of inter-areal dissimilarity across four cortical hierarchies. To examine differences between each cortical hierarchical level, two-sided two-sample *t*-tests were conducted with FDR correction (correlations with *: $p_{FDR} < 0.05$; correlations with **: $p_{FDR} < 0.001$). Sample sizes: Paralim (45), Hetero (75), Uni (79), and Idio (29); all are technical replicates. Box plots display the median (center line), interquartile range (box bounds = 25th to 75th percentiles), and whiskers extending to the minimum and maximum values within 1.5× the interquartile range (IQR). Bars are color-coded as follows: blue for paralimbic, orange for heteromodal, green for unimodal, and red for idiotypic. To explore associations with the histological gradient, two-sided Spearman's correlation coefficients were computed ($p = 0.001$), and *p* values corrected using spin permutation tests[62]. **C** The gradient profile of

parcel *i* was calculated by averaging the gradients across all vertices belonging to parcel *i*. The intra-areal dissimilarity of parcel *i* was then determined by calculating cosine distance between vertex-wise gradients and gradient profile of parcel *i*. By repeating this procedure for all parcels, we generated a map illustrating the distribution of intra-areal dissimilarity across the cortex. Regions with significantly higher intra-areal dissimilarity compared to other areas after applying FDR correction were highlighted using black boundaries ($p < 0.05$). **D** The distribution of vertex-wise intra-areal dissimilarity in four cortical hierarchies is shown in the left panel. To assess differences between each cortical hierarchy, two-sided two-sample *t*-tests were performed with FDR correction (correlations with *: $p_{FDR} < 0.05$). Sample sizes: Paralim (789), Hetero (2,342), Uni (2,403), and Idio (1779); all are technical replicates. Box plots display the median (center line), interquartile range (box bounds = 25th to 75th percentiles), and whiskers extending to the minimum and maximum values within 1.5× the IQR. To explore associations with the histological gradient, two-sided Spearman's correlation coefficient was computed with FDR correction. Source data are provided as a Source Data file. Abbreviation: Paralim Paralimbic, Hetero Heteromodal, Uni Unimodal, Idio Idiotypic.

The results revealed that the idiotypic system had the highest inter-areal dissimilarity compared to other systems (e.g., Unimodal vs Idiotypic: $t = -4.73$, $p_{spin} = 0.001$, Cohen's $d = -0.99$, 95% CI = [−0.068, −0.028], FDR correction, Fig. 2B). In contrast, paralimbic systems showed lowest inter-areal dissimilarity (e.g., Paralimbic-*vs*-Heteromodal: $t = -9.16$, $p_{spin} = 0.001$, Cohen's $d = -1.46$, 95% CI = [−0.089, −0.058], FDR correction), aligning with prior findings. To further explore associations between inter-areal dissimilarity and cortical microstructure, we generated a MPC matrix of histological data obtained from the BigBrain dataset[65], a 3D reconstruction of *post-mortem* human brain histology, and estimated its principal gradient. This gradient has previously been shown to closely recapitulate Mesulam's taxonomy of cortical hierarchical organization. In effect, we also observed a significant correlation between the histological gradient and inter-areal dissimilarity (rho = −0.61, $p_{spin} = 0.001$, 95% CI = [−0.706, −0.552]), supporting a close association between regional cytoarchitecture and macroscale organization.

To examine inter-areal similarities using an alternative approach, we performed hierarchical clustering on cortical similarity matrices and found equivalent results, providing four robust clusters recapitulating sensory-fugal hierarchies (Supplementary Fig. 1).

We furthermore explored the layout within each parcel to better understand local organization. Here, we calculated the cosine distance between the gradient profile of each cortical vertex and the mean gradient profile of the area to which it belongs, as a measure of intra-areal dissimilarity (Fig. 2C). We found that intra-areal dissimilarity was considerably lower compared to inter-areal dissimilarity (which also confirms the utility of the used parcellation). The medial supplementary motor areas exhibited highest intra-areal dissimilarity ($p_{spin} < 0.05$, spatial permutation tests, FDR correction), while the medial orbitofrontal cortex showed the lowest values under the same statistical threshold. Additionally, we found that intra-areal dissimilarity was positively correlated with parcel size (rho = 0.44, $p_{spin} = 0.001$). Comparing vertex-wise intra-areal dissimilarity across hierarchical levels, we did not observe the same relationship between different levels as for inter-areal dissimilarity. In fact, there was a trend indicating that intra-areal dissimilarity was higher in heteromodal and unimodal association systems compared to idiotypic and paralimbic regions, though this difference was only marginally significant (Paralimbic vs Heteromodal: $t = -6.03$, $p_{spin} = 0.042$, Cohen's $d = -0.25$, 95% CI = [−0.019, −0.010]; Paralimbic vs Unimodal: $t = -6.54$, $p_{spin} = 0.045$, Cohen's $d = -0.26$, 95% CI = [−0.021, −0.011], FDR correction; Fig. 2D). Moreover, there was no significant association to the histological gradient derived from BigBrain (rho = −0.17, $p_{spin} = 0.12$, 95% CI = [−0.280, −0.035]).

## Associations to functional diversity across task states

The cortical layout is intricately linked to cognition[23,66]. To investigate how functional connectivity patterns change across diverse cognitive states, we administered nine different fMRI tasks, including episodic memory encoding and retrieval, semantic retrieval, mnemonic similarity task (MST), and four passive movie watching paradigms in the same participants at 7 T. Functional connectivity for all cognitive states was constructed by cross-correlating the vertex-wise timeseries. We then calculated the cosine distance between the corresponding whole brain functional connectivity matrices to estimate cross-task diversity for each area (Supplementary Fig. 2). Focusing on overall diversity, we calculated the average of values across all tasks to generate the cross-task diversity map (Fig. 3A). We observed highest functional diversity in the medial temporal lobe and orbitofrontal cortex, while lower diversity in the medial frontal lobe and primary sensory cortex (Fig. 3B). To further explore how this functional diversity relates to cortical organization, we assessed associations between the cross-task diversity map and inter-areal dissimilarity. This was done by computing Spearman's correlation coefficient and correcting *p* values using 1000 spin permutation tests. Notably, we identified a marked correlation between cross-task diversity and inter-areal dissimilarity (rho = −0.71, $p_{spin} = 0.001$, 95% CI = [−0.738, −0.583]; Fig. 3B). To control for potential influences from image quality, we performed partial correlation analyzes. The results remained consistent when controlling for tSNR (rho = −0.46, $p_{spin} = 0.001$), B1+ field map (rho = −0.71, $p_{spin} = 0.001$), CNR from the T1 data (rho = −0.70, $p_{spin} = 0.001$), and SNR from the diffusion data (rho = −0.71, $p_{spin} = 0.001$). The results were consistent when we excluded regions with lowest tSNR (rho = −0.70, $p_{spin} = 0.001$). As expected, paralimbic areas exhibited more variable connectivity across different tasks, suggesting a more flexible functional integration into large-scale networks. In contrast, primary cortices showed less variable patterns, potentially supporting their more fixed functional specialization. Again, we only found a weak association with intra-areal dissimilarity (rho = −0.19, $p_{spin} = 0.065$, 95% CI = [−0.254, −0.056]).

We also investigated how intra-areal functional connectivity patterns changed across different task contexts (Fig. 3C). Here, we defined intra-areal cross-task diversity as the standard deviation between vertex-wise FC across different tasks, with the effect of parcel size controlled. We observed highest intra-areal diversity in the superior temporal lobe, while lower diversity was observed in paralimbic cortex. By computing Spearman's correlation coefficient, we identified a significant correlation between intra-areal diversity and inter-areal diversity (rho = −0.49, $p_{spin} = 0.001$, 95% CI = [−0.603, −0.407]). This correlation remained significant when controlling for tSNR as a covariate (rho = −0.48, $p_{spin} = 0.001$; Fig. 3C). No significant correlation

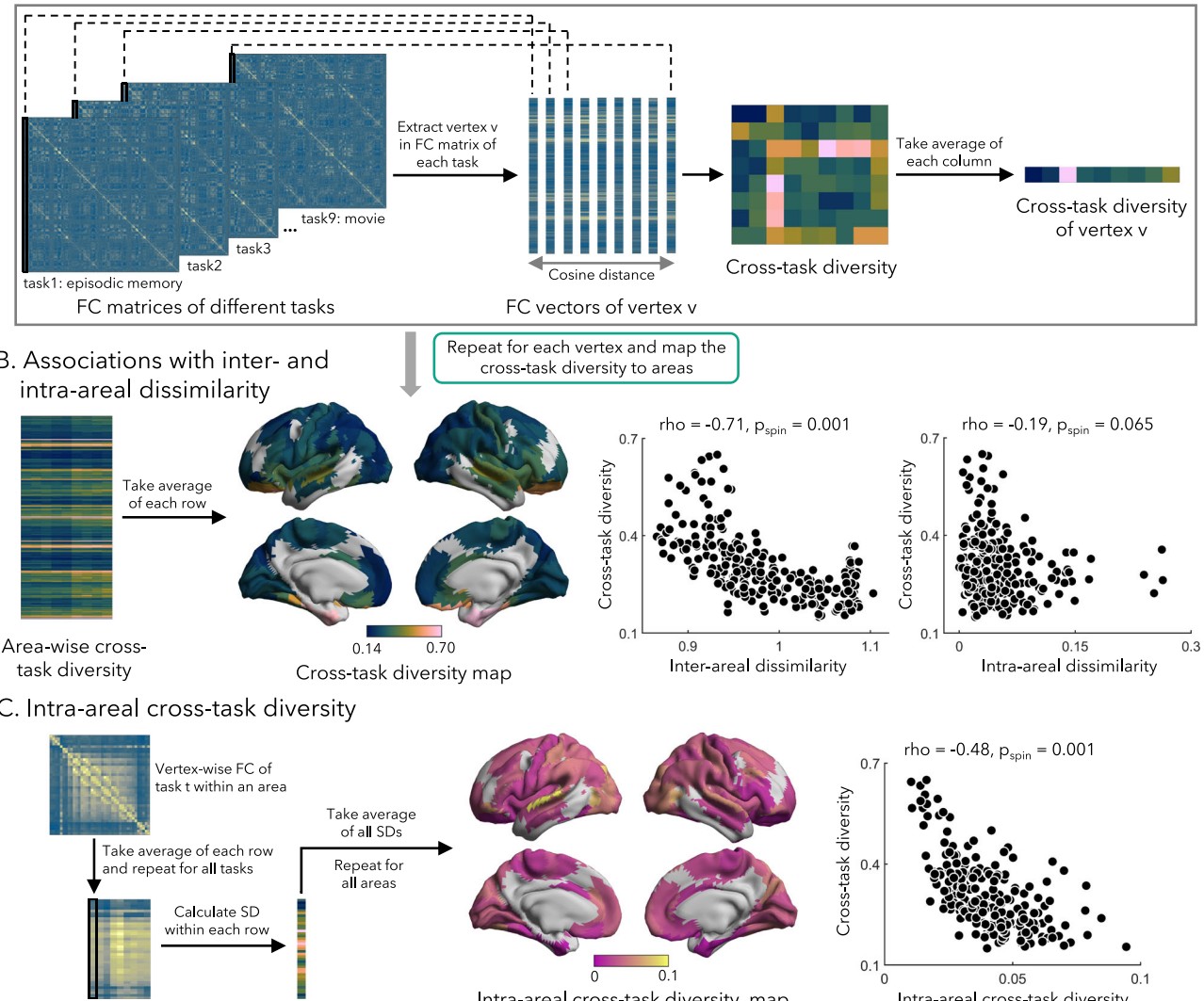

**Fig. 3 | Associations to cross-task functional diversity. A** The FC matrices were generated using time series data obtained from for nine tasks fMRI sessions in the same subjects. This was followed by the estimation of cosine distances across different tasks. This process resulted in a cross-task diversity matrix of dimensions. To quantify the cross-task diversity for the first vertex of all tasks, the average of each column in this matrix was computed. **B** By repeating the procedures outlined in (**A**) for each vertex and subsequently mapping the results to areas, we created an area-wise cross-task diversity map. Two-sided Spearman's correlation coefficient

was computed, and *p* values were corrected using spin permutation tests. **C** By taking the average of each row of the vertex-wise FC within area and calculating the standard deviation (SD) across tasks, we generated the vertex-wise cross-task SD. Taking the average of all SDs within an area and repeating this procedure for all areas, we generated the intra-areal cross-task diversity map. Two-sided Spearman's correlation coefficient was computed, and *p* values were corrected using spin permutation tests. Source data are provided as a Source Data file. Abbreviation: FC functional connectivity.

was observed between intra-areal diversity across tasks and intra-areal dissimilarity in gradient profiles, however.

### Robustness with respect to analysis parameters and parcellation atlas

To assess robustness of our findings with respect to analysis parameters, we recalculated gradients, gradient profiles, as well as inter- and intra-areal gradient profile dissimilarity for each modality using different thresholds of the connectivity matrix (50%, 60%, 70%, 80%, 90%). Correlations between inter- and intra-areal dissimilarity derived from gradient profiles with different thresholds were assessed, revealing consistent results (Supplementary Fig. 3). Moreover, we investigated the impact of varying the number of gradients within each modality, ranging from three to seven. Inter- and intra-areal dissimilarity was estimated, and consistent results were observed across

different numbers (Supplementary Fig. 3). To assess the effect of using different parcellation atlases, the main results were replicated using the Glasser atlas[3]. This atlas is purely MRI-based, but aggregates information from different modalities and offers whole-cortex coverage. Again, consistent results were observed when using this atlas (Supplementary Fig. 4).

### Reliability at the single-subject level

We assessed our findings at each of the ten individual participants who were scanned at 7 T. Similar results were found across all participants, including gradient profile matrices (Fig. 4A), and measures in inter- and intra-areal dissimilarity (Supplementary Fig. 5). Moreover, we observed marked negative correlations between inter-areal dissimilarity and the histological gradient (rho = −0.58 ± 0.047, ranged from −0.63 to −0.49, all $p_{spin} < 0.001$), and between inter-areal dissimilarity and cross-task

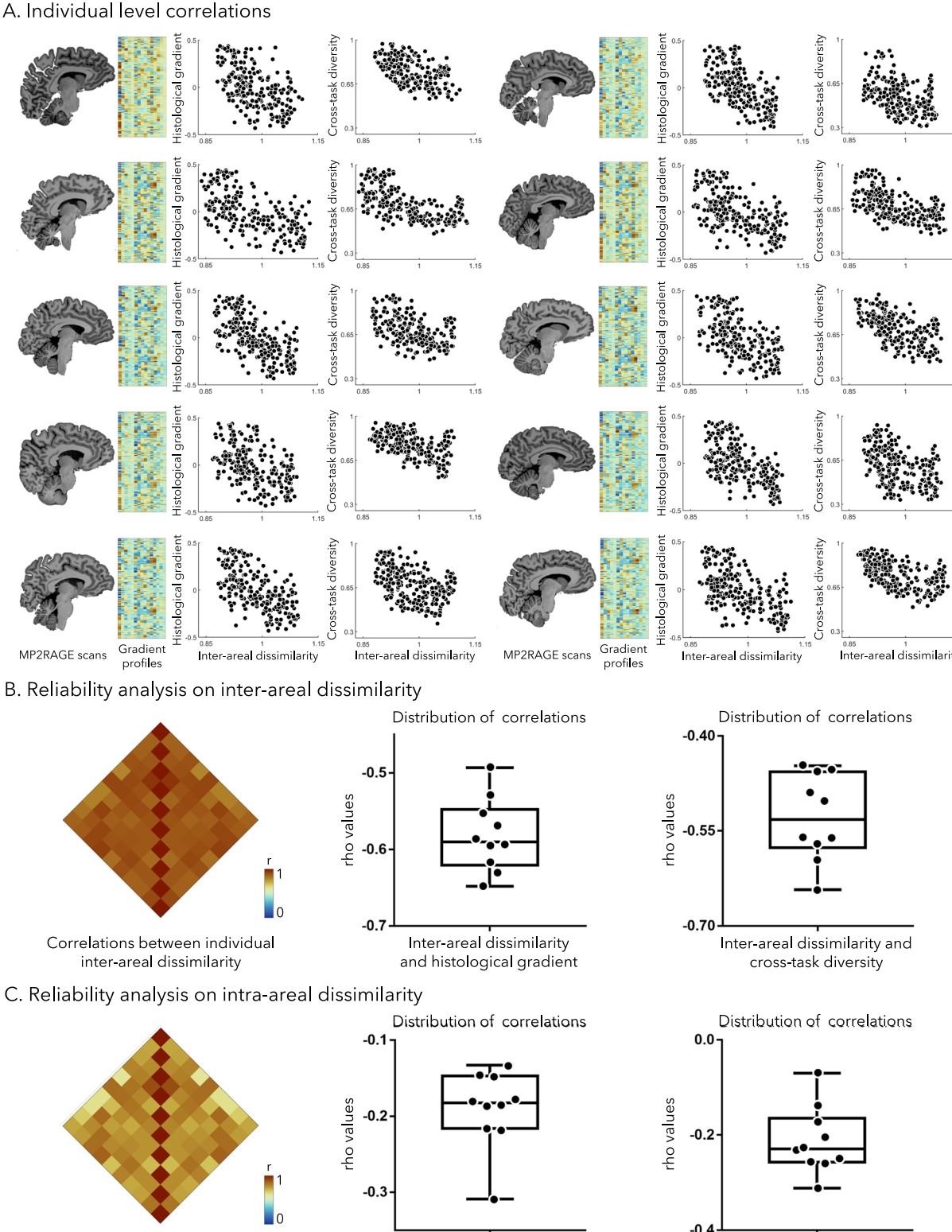

A. Individual level correlations

B. Reliability analysis on inter-areal dissimilarity

Correlations between individual inter-areal dissimilarity

Distribution of correlations — Inter-areal dissimilarity and histological gradient

Distribution of correlations — Inter-areal dissimilarity and cross-task diversity

C. Reliability analysis on intra-areal dissimilarity

Correlations between individual intra-areal dissimilarity

Distribution of correlations — Intra-areal dissimilarity and histological gradient

Distribution of correlations — Intra-areal dissimilarity and cross-task diversity

diversity (rho = −0.53 ± 0.067, ranged from −0.64 to −0.45, all $p_{\mathrm{spin}} < 0.001$). Inter-areal dissimilarity across participants was highly correlated ($r = 0.90 ± 0.008$; Fig. 4B). Also, intra-areal dissimilarity across participants was highly correlated ($r = 0.76 ± 0.007$; Fig. 4C). Individual-level intra-areal dissimilarity was considerably lower compared to inter-areal dissimilarity, and no significant associations to the histological gradient or cross-task diversity were observed. To

examine the effect of sex, we conducted sex-disaggregated analyzes on the main results and observed consistent findings across both male and female subgroups.

### Replication at 7 T
We conducted replication analysis based on 12 healthy young adults (age: 29.75 ± 4.75 years, 7 females) with one session scanned at 7 T. We

**Fig. 4 | Reliability analysis at the single subject level. A** For each of the 10 subjects, we generated gradient profiles and inter-/intra-areal dissimilarity. Associations between individual inter-areal dissimilarity and histological gradient, as well as cross-task diversity were examined using Spearman's correlation coefficients. **B** Associations between individual inter-areal dissimilarity were estimated using two-sided Pearson's correlation. Distributions of rho values between inter-areal dissimilarity and histological gradient (mean ± SD of rho values = −0.58 ± 0.047; sample sizes $n = 10$, with 10 biological replicates), and with cross-task diversity were examined (mean ± SD of rho values = −0.53 ± 0.067; sample sizes $n = 10$, with 10 biological replicates). Box plots display the mean (center line = 50% of the group), interquartile range (box bounds = 25th to 75th percentiles), and whiskers extending to the minimum and maximum values within 1.5× the inter-quartile range (IQR). **C** Associations between individual intra-areal dissimilarity were estimated using two-sided Pearson's correlation. Distributions of rho values between inter-areal dissimilarity and histological gradient (mean ± SD of rho values = −0.19 ± 0.051; $n = 10$, with 10 biological replicates), and with cross-task diversity were examined (mean ± SD of rho values = −0.21 ± 0.069; $n = 10$, with 10 biological replicates). Box plots display the mean (center line = 50% of the group), interquartile range (box bounds = 25th to 75th percentiles), and whiskers extending to the minimum and maximum values within 1.5× the IQR. Abbreviation: MP2RAGE magnetization-prepared 2 rapid gradient echo. Source data are provided as a Source Data file.

observed consistent results as for the main findings (Supplementary Fig. 6), supporting reproducibility.

## Replication at 3 T

We conducted a second replication analysis in an independent sample of 100 healthy adults (age: 34.33 ± 4.07 years, 47 females) scanned at 3 T[61]. Results were consistent, with similar multimodal gradients, gradient profiles, and inter-areal dissimilarity (Supplementary Fig. 7A, B). Notably, associations with histological gradients (rho = −0.40, $p_{spin} < 0.001$) and cross-task diversity (rho = −0.59, $p_{spin} < 0.001$) remained consistent, albeit with slightly smaller effects as for the 7 T dataset. Again, we observed only a marginal association between intra-areal dissimilarity and cross-task diversity (rho = −0.09, $p_{spin} > 0.1$; Supplementary Fig. 7C), with no significant correlation found for histological gradients.

## Discussion

Functional specialization and integration are two cornerstones of neural organization[67,68]. While specialization relates to distinctive neural behavior across different contexts[69], functional integration emphasizes the shared influence among regions, ultimately contributing to coherent experiences and behavior[69,70]. The present study combined multimodal MRI acquisitions with robust areal descriptions of cortical cytoarchitecture[19], in order to identify the similarity and divergence of inter-areal gradient fingerprints. Vertex-wise multimodal connectomes were constructed from high-field 7 T MRI data, and cortical gradients were estimated, aligning with those described in prior studies[22,23,59]. We noted higher inter-areal dissimilarity in sensorimotor cortices and lower inter-areal dissimilarity in the transmodal system, indicating the distinctiveness of the primary sensory cortex as a functionally specialized system. Additionally, functional connectivity in the primary cortex exhibited less variability across tasks, suggesting that information processing from different tasks may converge at early stages. Significant associations were identified between cross-task functional diversity and inter-areal gradient profile dissimilarity, indicating a link between global cortical motifs and functional flexibility across different task contexts. These findings suggest a sensory-paralimbic differentiation in cortical gradient profiles, providing insights into neural motifs contributing to specialized and integrative brain function.

The availability of multimodal neuroimaging data offers opportunities for examining brain organization across different spatial scales[12,41,42]. In this work, we leveraged repeated MRI scanning at ultra-high fields of 7 T, which we hypothesized would result in high signal[71], reliability, and precision[53–57]. The first principal component of our estimated multimodal gradient profiles demonstrated that sensory-functional axes jointly guide hierarchical patterns of cortical microstructure, structure, and function. Our work presents a converging overarching principle, extending prior work that has focused on specific modalities in isolation[22,23,72]. These findings suggested a convergence of organizational principles across different scales and at the level of structure and function. Notably, and in addition to harnessing a multimodal imaging approach, we leveraged a recently disseminated atlas derived from post-mortem histological data to finely partition the brain into distinct areas based on cytoarchitecture, a presumed gold standard for cortical mapping[19]. Our work, thereby, integrated dimensional gradient and area-based descriptions of macroscale cortical spatial patterns, providing a unified framework for understanding the cortical layout. This framework could identify key principles of between- and within-parcel heterogeneity, distinguish different zones across the cortical hierarchy, and reveal associations between these heterogeneity patterns and cross-task diversity. Given that the gradient profiles in our study represent vectors in a multidimensional space, we then measured the distance between all cortical areas using a cosine distance metric. As expected, we observed an overarching pattern of inter-areal dissimilarity, with one end featuring sensory and motor cortices that had the highest dissimilarity and the other end encompassing heteromodal and paralimbic areas in the transmodal apex. This suggests that functional specialization in primary sensorimotor areas is accompanied by a more distinctive organization pattern. Reliability analyzes indicated that this inter-areal pattern was not influenced by potential methodological biases, including the B1+ field map, SNR of diffusion data, or CNR of T1 data. While tSNR was correlated with inter-areal dissimilarity, the correlation between inter-areal dissimilarity and cross-task diversity remained significant even after controlling for tSNR. Overall, the axis of inter-areal dissimilarity is consistent with the gradients previously reported for single modalities that are, especially at the level of function and microstructure, also anchored in primary systems on the one end, and heteromodal and paralimbic regions on the other end[22,23]. Tract-tracing and neuroimaging experiments have documented that primary sensory and motor cortices host more short-range cortico-cortical connections than transmodal systems[60], and these regions also tend to have a higher coupling between microstructure and function, and between structural connectivity and function[27]. Such findings are potentially in support of their more specialized functional profiles[60,73]. Notably, the pattern of cortical organization within the limbic system closely aligns with that of other cortical areas, extending previous findings derived from single modalities[22,23,72,73]. For instance, the paralimbic system with more long-distance connections was reported to have higher microstructural similarity[60]. Similarly, another study combining structural and functional connections revealed that less diverse unimodal regions show a preference for local communication, while more diverse multimodal regions engage in more global communication patterns[74]. Indeed, the degree of laminar differentiation also varies gradually across the cortex, reflecting the specialization of underlying cortical microcircuits[75,76]. It is highest in primary areas, such as the visual and somatosensory cortices, and then decreases in unimodal and heteromodal regions, to reach its lowest level in agranular areas[7,77]. Projection patterns similarly follow this gradient[78–80]. Together, these organizational patterns, spanning from the microscale to the macroscale, may support the integrative role of the limbic system, enabling it to participate in various cognitive processes[81]. However, it is important to note that no significant associations were observed between PC[63]

and either inter-areal dissimilarity or cross-task functional diversity, suggesting a more complex relationship between cortical heterogeneity, functional diversity, and different indices of macroscale functional integration.

Distance-dependence theory suggests that proximal regions are more likely to be inter-connected[82,83]. In addition, those areas sharing similar microstructural and neurobiological characteristics are also more likely to be interconnected than regions with distinct features[78,84,85]. These findings support the overarching idea that adjacent regions with short-range connections often share gene expression and microstructural similarities, contributing to their specialized functional roles. Conversely, while nearby neurons are expected to share similar microstructural properties when extending smooth macroscale topography to the microscale[86], prior electrophysiological experiments in various mammalian brain regions have shown that nearby neurons can exhibit disparate response properties[87–90]. By examining intra-areal dissimilarity in gradient motifs in the current study, we furthermore probed intra-regional consistency vs heterogeneity in neural organization. Overall, intra-areal gradient profile divergence was relatively low compared to inter-areal divergences, confirming the utility of areal parcellations to meaningfully subdivide the cortex more generally[56,91]. Notably, although there was no clear difference in intra-areal dissimilarity between networks, we observed a trend towards increased dissimilarity within the unimodal and heteromodal cortices. We speculate that this may be related to their more integrative functional role and increased functional flexibility compared to primary sensory and motor areas, which deserves further verification in future work. Meanwhile, the potential effects of region size should also be considered, as larger regions are more likely to exhibit higher intra-region dissimilarity. Moreover, and in contrast to the inter-areal findings, we did not observe noteworthy associations between intra-areal variations and functional and microstructural indices of the cortical hierarchy. We speculate that intra-regional heterogeneity can support the processing and integration of inputs across a given cortical territory[76]. That is, for disparate types of information to be integrated, they must at minimum be present in the same brain regions. Moreover, these findings are aligned with the tethering hypothesis of cortical patterning, where a disproportionate enlargement of uni- and heteromodal systems during human evolution and the progressive decrease of genetically mediated signaling gradients may have contributed to their higher intra-areal dissimilarity and relative structure-function decoupling[30,92].

There is growing evidence for flexibility in the functional organization of the cortex, even within a relatively fixed structural and cytoarchitectural layout[38–40,93]. For instance, recent studies found that functional activation and connectivity change in certain areas in the same individual across different tasks[38,39,93]. Another study found that graph theory measures such as clustering coefficient and nodal degree can change significantly when comparing the same subjects across different task conditions[40]. To explore this further, we investigated the inter and intra-areal functional diversity across different fMRI tasks in the same participants. Multiple patterns of inter-areal functional diversity were found in association cortices, consistent with prior studies[39], where cortical regions specialized for the same functions were strongly coupled. However, high diversity was found in functionally flexible regions that participated in multiple functions, supporting their role in integrating specialized brain networks[39]. We also found a negative correlation between cross-task functional diversity and inter-areal dissimilarity, suggesting that globally more specialized regions, such as sensorimotor and visual cortices, exhibit more stable functional connectivity patterns across tasks. A recent study reported that functional connectivity in somatomotor cortex increased with age during childhood through adolescence, whereas it declined in association cortices, reinforcing the differentiation of sensorimotor and association systems in typical development[94]. These findings support the existence of a sensorimotor-association axis of cortical organization[8,22], and may explain higher stability of functional connectivity in sensorimotor cortex across tasks. For intra-areal cross-task diversity, we identified a specific axis, indicating relatively stable patterns in primary sensory and paralimbic cortices. Interestingly, paralimbic areas exhibited the least intra-areal dissimilarity compared to other cortical regions, showing higher inter-areal diversity but lower intra-areal diversity in functional connectivity across tasks. This may be due to the involvement of these regions in a wide range of cognitive and affective processes[81,95]. Prior findings suggest that paralimbic cortices with a simple laminar structure are well suited to integrate a neural "workspace" for a unified conscious experience due to their position in cortical hierarchies and their connectivity[96]. We inferred that the low intra-areal diversity in paralimbic cortices across tasks is due to their relatively simple local connectivity structure. However, the interaction patterns between the paralimbic cortices and other brain regions are dynamic, adapting to integrate different sensory inputs and meet various cognitive demands when performing different tasks[96,97]. Collectively, these findings suggest that the heterogeneity in global and local cortical motifs across different regions is reflected in their diverse participation across different functional contexts. A series of robustness analyzes, exploring the influence of thresholds for gradient estimation and the number of gradients, yielded similar results, suggesting that our analyzes was not affected by variations in specific analysis parameters. Moreover, we observed consistent findings at the level of individual participants and could replicate our findings using an independent dataset of healthy participants scanned at 3 T. Nevertheless, future studies could explore additional sources of variability, both between participants and within the same participant over time, which may arise from individual differences and temporal factors[98]. In this study, we focus on structure-function relationships within areas with similar cytoarchitecture and how these relationships differ between areas. Using the Julich-Brain atlas offers significant benefits because it is based on ground-truth cytoarchitecture. However, a potential limitation is its lack of coverage of the entire cortex. This limitation is anticipated to be addressed with the future publication of the whole-brain probabilistic map, currently under development[19]. To mitigate this issue in the current work, we replicated our main analyzes using the Glasser atlas and found consistent results. As our work shows, cortical parcellation and gradient descriptions provide synergistic information to understand human brain organization. By thus reconciling local and global cortical patterns, our work provides insights into the neuroanatomical basis of specialized and integrative cortical functions.

## Methods

### Participants

Our study was based on three independent human neuroimaging datasets. A 7 T dataset (MICA-PNI, 10 subjects, multiple time points) was for the main analysis and cross-subject reliability assessment. A 7 T dataset (*MICA-7T*, 12 subjects, one time point) was used for replication analysis. A 3 T dataset (MICA-MICs, 100 subjects, one time point) was used for replication. Sex was determined through self-reporting by participants and was considered in the study design. To minimize potential biases, we aimed to collect an equal amount of data from both male and female participants.

**MICA-PNI.** For our main analysis, we investigated the imaging and phenotypic data of 10 unrelated healthy adults (age: 29.20 ± 5.20 years, 5 females). Each participant underwent three sessions on separate days. Data were collected between March 2022 and June 2023. This dataset is openly available at the OSF platform (https://osf.io/mhq3f/).

**MICA-7T**. This dataset consisted of 12 unrelated healthy young adults (age: 29.75 ± 4.75 years, 7 females). Data were collected between May 2023 and May 2024.

**MICA-MICs[61]**. This dataset consisted of 100 unrelated healthy young adults (age: 34.33 ± 4.07 years, 47 females). Data were collected between April 2018 and February 2021. A subset of 50 participants is openly available (https://portal.conp.ca/dataset?id=projects/mica-mics)[61].

The studies were approved by the Ethics Committees of McGill University and the Montreal Neurological Institute and Hospital, respectively, and written and informed consent were obtained from all participants. In addition, participants were compensated financially for each MRI scanning session attended as reimbursement for their time and participation.

## MRI acquisition

**MICA-PNI**. MP2RAGE is acquires two 3D images with different inversion times (TI) to generate a myelin-sensitive map of the T1 relaxation times and a synthetic T1-weighted (T1w) image. The 3D MP2RAGE sequence parameters are the following: 0.5 mm isovoxels, matrix = 320 × 320, 320 sagittal slices, repetition time (TR) = 5170 ms, echo time (TE) = 2.44 ms, TI = 900 ms, flip angle = 4°, iPAT = 3, TI1 = 1000 ms, TI2 = 3200 ms, bandwidth = 210 Hz/px, echo spacing = 7.8 ms, and partial Fourier = 6/8. Scans were visually inspected to ensure minimal head motion, and repeated if necessary. Both inversion images were combined for T1 mapping and to minimize sensitivity to B1 inhomogeneities[99,100].

DWI data was acquired using a multiband accelerated 2D spin-echo echo-planar imaging sequence. The acquisition included three shells with b-values of 300, 700, and 2000 s/mm2, and 10, 40, and 90 diffusion weighting directions per shell, respectively. The parameters used were: 1.1 mm isotropic voxels, TR = 7383 ms, TE = 70.60 ms, flip angle = 90°, refocusing flip angle = 180°, FOV = 224 × 224 mm², slice thickness = 1.1 mm, multi-band factor = 2. Reverse phase encoding b0 images were obtained for distortion correction of the DWI scans.

All multi-echo fMRI were acquired with a 2D blood oxygenation level dependent (BOLD) echo-planar imaging sequence. The parameters were as follows: 1.9 mm isotropic voxels, TR = 1690 ms, TE1 = 10.8 ms, TE2 = 27.3 ms, TE3 = 43.8 ms, flip angle = 67°, FOV = 224 × 224 mm², slice thickness = 1.9 mm, multiband factor = 3, and echo spacing = 0.53 ms. During the 6-min rs-fMRI scan, participants were instructed to keep their eyes open, fixate on a cross presented on the screen, and not think of anything. Two spin-echo images with opposite phase encoding directions were also acquired for distortion correction of the rs-fMRI scans, with the following parameters: phase encoding = AP/PA, 1.9 mm isovoxels, FOV = 224 × 224 mm², slice thickness = 1.9 mm, TR = 3000 ms, TE = 18.4 ms, flip angle = 90°. Based on a validated open-source protocol[101], we collected multiple task fMRI scans, including episodic encoding/retrieval and semantic tasks, as well as the MST, which both lasted ~6 min. During the episodic memory encoding, participants memorized paired images of objects. In the retrieval phase, participants were shown an image and asked to identify the paired object from three options. Semantic memory retrieval involved identifying the object that is most conceptually related to a target image from three options. In all memory tasks, there were 56 trials, and the difficulty was modulated based on semantic relatedness scores[102]. During the MST, participants determined whether the object in images was indoor or outdoor, and then identified whether the object shown in images was old, similar, or new. We also collected fMRI data while participants watched movies, tracking hemodynamic activity during naturalistic viewing conditions[103]. A detailed imaging protocol is provided in the data release (https://osf.io/mhq3f/), including the complete list of acquisition parameters.

**MICA-MICs**. Two T1w scans with identical parameters were acquired with a 3D magnetization-prepared rapid gradient echo sequence (0.8 mm isovoxels, matrix = 320 × 320, 224 sagittal slices, TR = 2300 ms, TE = 3.14 ms, TI = 900 ms, flip angle = 9°, iPAT = 2). Scans were visually examined to ensure minimal head motion, and repeated if necessary.

qT1 relaxometry data was acquired using a 3D-MP2RAGE sequence (0.8 mm isovoxels, 240 sagittal slices, TR = 5000 ms, TE = 2.9 ms, TI 1 = 940 ms, T1 2 = 2830 ms, flip angle = 4°, flip angle 2 = 5°, iPAT = 3, bandwidth = 270 Hz/px, echo spacing = 7.2 ms, partial Fourier = 6/8). Both inversion images were combined for qT1 mapping to minimize sensitivity to B1 inhomogeneities and optimize intra- and inter-subject reliability[99,100].

DWI data was acquired using a 2D spin-echo echo-planar imaging sequence, consisting of three shells with b-values = 300, 700, and 2000 s/mm², and with 10, 40, and 90 diffusion weighting directions per shell, respectively (1.6 mm isovoxels, TR = 3500 ms, TE = 64.40 ms, flip angle = 90°, refocusing flip angle = 180°, FOV = 224 × 224 mm², slice thickness = 1.6 mm, multi-band factor = 3, echo spacing = 0.76 ms). b0 images acquired in reverse phase encoding direction were used for distortion correction of DWI scans.

A 7-min rs-fMRI scan was acquired using multiband accelerated 2D-BOLD echo-planar imaging (3 mm isovoxels, TR = 600 ms, TE = 30 ms, flip angle = 52°, FOV = 240 × 240 mm², slice thickness = 3 mm, mb factor = 6, echo spacing = 0.54 ms). Participants were instructed to keep their eyes open, not fall asleep, and look at a fixation cross. Two spin-echo images with reverse phase encoding were also included for distortion correction of the rs-fMRI scans (phase encoding = AP/PA, 3 mm isovoxels, TR = 4029 ms, TE = 48 ms, flip angle = 90°, FOV = 240 × 240 mm², slice thickness = 3 mm, echo spacing = 0.54 ms, bandwidth = 2084 Hz/Px).

## Multimodal MRI processing

**MICA-PNI**. The MP2RAGE scans of each subject were reoriented using FSL[104], linearly co-registered, averaged, with background noise removed, corrected for intensity nonuniformity using N4 bias field correction from ANTS[105], and segmented into white and gray matter using FSL FAST[104]. Resulting volumes were skull stripped using FSL[104,106]. Cortical surface models were generated from native T1w scans using FastSurfer[107]. Surface reconstructions for each subject underwent manual correction for segmentation errors, by placing control points and applying manual edits. The B1+ field map and the CNR of the T1 map were calculated for each participant to assess reliability.

Regarding the DWI data, pre-processing was carried out using MRtrix[108] in the native DWI space. The DWI data underwent denoising[109,110], b0 intensity normalization, and correction for susceptibility distortion, head motion, and eddy currents. These corrections were performed using FSL[111] and involved utilizing two $b = 0$ s/mm² volumes with reverse phase encoding. Anatomical masks for tractography were non-linearly co-registered to native DWI space using the deformable SyN approach implemented in ANTs[112]. We computed the mean and standard deviation maps of the b0 images. For each participant, voxel-wise SNR maps were generated by dividing the mean map by the standard deviation map. These SNR maps were subsequently projected onto the fsLR-5k surface.

For the rs-fMRI scans, pre-processing steps were conducted using AFNI[113] and FSL[104] tools. The first five volumes were discarded to ensure magnetic field saturation. We applied Multi-Echo Independent Components Analysis[114,115] to improve the signal-to-noise ratio and effect of motion correction. Spike regression was applied to remove timepoints with large motion spikes, effectively removing nuisance signals[116,117]. The volume time series were registered to FastSurfer[107] space using boundary-based registration implemented in ANTs using linear and

non-linear methods[118]. The tSNR maps were estimated for each participant.

**MICA-MICs.** The surface reconstructions for each subject underwent manual inspection and correction for segmentation errors by placing control points and applying manual edits. The qT1 scans were linearly co-registered to the corresponding subject's T1w scan.

Regarding the DWI data, pre-processing was carried out using MRtrix[108] in the native DWI space. The DWI data underwent denoising[109,110], b0 intensity normalization, and correction for susceptibility distortion, head motion, and eddy currents. These corrections were performed using FSL and involved utilizing two $b = 0 \, s/mm^2$ volumes with reverse phase encoding. Anatomical masks for tractography were non-linearly co-registered to native DWI space using the deformable SyN approach implemented in ANTs[112].

For the rs-fMRI scans, pre-processing steps were conducted using AFNI[113] and FSL[104] tools. The first five volumes were discarded to ensure magnetic field saturation. The remaining volumes underwent reorientation, motion correction, and distortion correction. We applied FMRIB's ICA-based X-noiseifier[119] and spike regression to remove timepoints with large motion spikes, effectively removing nuisance signals[116,117]. The volume time series were registered to FastSurfer[107] space using boundary-based registration implemented in ANTs using linear and non-linear methods[118].

### Generating multimodal connectome matrices

To investigate the vertex-wise multimodal connectomes, we first constructed a downsampled fsLR-5k surface using HCP's workbench tools (wb_command)[120]. The fsLR-32k surface templates and resampling spheres between "fsaverage" and "fs_LR" were accessed from the HCP's open-access pipeline[121]. Subsequently, we downsampled the surface template, registration spheres, and mid-wall mask to 5k, resulting in a mesh comprising 4432 cortical vertices for each hemisphere. All vertex-wise analyzes were performed based on this fsLR-5k surface.

We calculated vertex-wise MPC matrices for each participant. Consistent with previous work[23,122,123], we constructed 14 equivolumetric surfaces between the pial and white matter boundaries to sample qT1 intensities across cortical depths. This procedure generated distinct intensity profiles reflecting intracortical microstructural composition at each cortical vertex. Data sampled from surfaces closest to the pial and white matter boundaries were removed to mitigate partial volume effects. Intensity values at each depth were mapped to a common template surface, resampled to fsLR-5k surface, and spatially smoothed across each surface independently (full width at half maximum [FWHM] = 3 mm). Vertex-wise intensity profiles were cross-correlated using partial correlations controlling for the average cortexwide intensity profile and log-transformed. This procedure resulted in the MPC matrices representing participant-specific similarity in myelin proxies across the cortex.

To generate each individual's SC, we employed MRtrix on preprocessed DWI data[108]. Each surface vertex from the fsLR-5k surface was translated into a volumetric region of interest that filled the cortical ribbon using workbench tools[120]. This process yielded ~10 k seeds/targets for structural connectome generation. Anatomical segmentations and volumetric seeds were then mapped to DWI space, applying the non-linear registration warp-field mentioned earlier. Next, we estimated multi-shell and multi-tissue response functions[124] and performed constrained spherical deconvolution to derive a fiber orientation distribution map[125,126]. This procedure, achieved through MRtrix, generated a tractogram with 40 M streamlines, with a maximum tract length of 250 mm and a fractional anisotropy cutoff of 0.06. To reconstruct whole-brain streamlines weighted by cross-sectional multipliers[127], we applied spherical deconvolution informed filtering of tractograms (SIFT2). Connection weights between seeds/targets were defined as the streamline count after SIFT2.

Next, individual rs-fMRI timeseries mapped to subject-specific surface models were resampled to fsLR-5k surface. Surface-based rf-MRI data underwent spatial smoothing with a Gaussian kernel (FWHM = 3 mm). An individual's FC matrix was generated by cross-correlating all vertex-wise timeseries. Correlation values subsequently underwent Fisher-R-to-Z transformations. FC matrices of all task fMRI scans were also generated using the same approach.

### Construction of gradient profiles

We converted each participant's SC, MPC, and FC matrices to a normalized angle affinity matrix, respectively, and applied diffusion map embedding on these matrices to generate multimodal gradients[128]. This non-linear dimensionality reduction procedure identified eigenvectors that describe main spatial axes of variance. Procrustes analysis aligned subject-level gradients to a group-level template generated from the group-average matrix of all participants. Gradients of the right hemisphere were aligned to the left hemisphere. Gradient analyzes were performed using BrainSpace (v0.1.10; http://github.com/MICA-MNI/BrainSpace), limiting the number of gradients to 10 and using default sparsity (keeping the top 10% of SC weights) and diffusion ($\alpha = 0.5$) parameters[129]. Here, we focused on the first five principal gradients of each modality (Fig. 1A). For each modality, all gradients were normalized by dividing by the maximum value in the absolute value of gradients, with values ranging from −1.0 to 1.0.

The Julich-Brain is a 3D probabilistic atlas of the human brain's cytoarchitecture, resulting from the analysis of 10 *post-mortem* human brains[19] (Fig. 1B). The probabilistic cytoarchitectonic maps (Julich-Brain v2.9, https://julich-brain-atlas.de/) were projected onto a template fsLR-5k surface to generate a surface-based representation[19]. Surface-based probabilistic maps contained values indicating the probability of an area being localized in each voxel, ranging from 0% to 100% overlap, with values ranging from 0 to 1. We registered the probabilistic maps to the fsLR-5k surface template. For each vertex, we defined its area label by identifying the area with the highest probability at that position. This area label was then used to assign all vertices on the fsLR-5k surface to the 228 areas defined by the Julich-Brain.

### Gradient profile analyzes

**Inter-areal heterogeneity and homogeneity assessment.** Understanding the relationships between diverse brain regions, including their similarities and differences, is essential for investigating the spatial patterns of brain organization. In this study, we focus on exploring inter-areal heterogeneity and homogeneity to further reveal the global layout of cortical area. To quantify inter-areal heterogeneity, we computed the inter-areal dissimilarity for each area. Specifically, we calculated the cosine distance between gradient profiles of each area, resulting in a cosine distance matrix (Fig. 2A). The inter-areal dissimilarity was defined as the mean of each row in the cosine distance matrix, representing the distance between an area and all other areas. To identify cortical regions with significantly higher or lower inter-areal dissimilarity, we projected the inter-areal dissimilarity map onto a sphere and conducted 1000 spin permutation tests. An area was considered to have the highest inter-areal dissimilarity among the cortex if its original inter-areal dissimilarity value exceeded 97.5% of the permutation values. Conversely, an area was regarded as having the lowest inter-areal dissimilarity if its original value was lower than 97.5% of the permutation values. To correct for multiple comparisons, we applied the FDR correction. To further investigate patterns of inter-areal dissimilarity, we conducted a network-level analysis utilizing the scheme proposed by Mesulam[64], which delineates four cortical functional zones (i.e., idiotypic, paralimbic, unimodal, heteromodal; see Fig. 2B). Inter-areal dissimilarity between each pair of cortical

hierarchies was compared using a two-sample *t*-test. FDR corrections were applied to correct for multiple comparisons, while spatial auto-correlation spin permutation tests were conducted for all tests. In order to explore the associations between inter-areal dissimilarity and cortical microstructural hierarchy, we generated a MPC matrix based on histological data from BigBrain[65], an ultrahigh-resolution 3D human brain model. From this matrix, we estimated the principal histological gradient as a representation of microstructural hierarchy. To examine the associations between the histological gradient and inter-areal dis-similarity, we calculated Spearman's correlation coefficient, with p-values corrected using 1000 spin permutation tests. To investigate the relationship between inter-areal dissimilarity and functional inte-gration patterns, we calculated the PC[63] using four cortical hierarchical levels[64] as communities. Additionally, we estimated global FC strength by averaging the FC values for each region.

Differences between cortical regions are crucial to functional specialization, but at the same time, similarities between regions support the realization of higher-order cognitive functions and func-tional integration across brain regions. To assess the homogeneity of cortical areas, we calculated (1-cosine distance) to represent the simi-larity between regional gradient profiles, resulting in an affinity matrix (Supplementary Fig. 1A). To evaluate the association between inter-areal similarity and cortical hierarchy, we examined the distribution of similarity coefficients across four hierarchy levels. To identify cortical areas with higher similarity, we performed hierarchical clustering on the affinity matrix to detect groups among the areas. We evaluated the clustering performance by calculating criterion values to determine the optimal number of clusters. We scrutinized the clusters with the highest criterion value and assessed the distribution of cortical hier-archies within each cluster to investigate the association between inter-areal homogeneity and cortical laminar differentiation.

**Intra-areal homogeneity and heterogeneity assessment.** Functional segregation of distinct regions is a critical principle of the human brain. It is thus important to investigate the local organization, i.e., the layout within an area to provide insights into the wiring principle of the cortex. Given that each area was originally defined based on shared neuroanatomical features, we expect to find overall high intra-areal homogeneity. Here we quantified, however, to what extent the level of homogeneity varies across the brain. We assessed inter-areal dissim-ilarity at both the region-level and network-level. As previously described, we calculated the gradient profiles for each area by aver-aging the vertex-wise gradients within that area. For a given area *i*, we calculated the cosine distance between the vertex-wise multimodal gradients and gradient profile of area *i*, resulting in the generation of intra-areal dissimilarity of area *i* (Fig. 2C). This procedure was repeated for all areas, yielding the vertex-wise intra-areal dissimilarity map.

To visualize the patterns of intra-areal heterogeneity more effec-tively, we calculated the average vertex-wise intra-areal dissimilarity within each area. By controlling for the number of vertices within each area, we accounted for the effect of area size. Furthermore, we inves-tigated the intra-areal dissimilarity at the network-level using the four cortical hierarchies proposed in a previous study[64]. We examined the distribution of intra-areal dissimilarity within these four cortical hier-archies and compared the differences between each hierarchy using a two-sample *t*-test with FDR correction and spatial autocorrelation spin permutation tests. To investigate associations between intra-areal dissimilarity and cortical microstructural hierarchy, we calculated Spearman's correlation coefficient between the histological gradient and intra-areal dissimilarity, correcting the *p* value using 1000 spin permutation tests.

**Associations to cross-task functional diversity.** To explore patterns of functional diversity across tasks, we constructed vertex-wise FC matrices by cross-correlating timeseries data derived from multiple task fMRI sessions. The cosine distance between each vertex from different tasks was then computed, resulting in a cross-task diversity matrix (Fig. 3A). To quantify the cross-task diversity for a specific task, we averaged all distance values between that task and others. This process was repeated for all tasks, and the outcomes were mapped to areas, generating area-wise cross-task diversity for each task. Focusing on overall diversity, we calculated the average of values across all tasks to create the cross-task diversity map (Fig. 3B). To investigate asso-ciations between cross-task diversity and inter-/intra-areal dissim-ilarity, we computed Spearman's correlation coefficient. The resulting *p* values were corrected for spatial autocorrelations using 1000 spin permutation tests.

To further investigate how intra-areal FC patterns change with tasks, we estimated the intra-areal cross-task diversity. For each area, we calculated the mean of FC strength between each vertex *v* to other vertices within a. We repeated this process for all tasks *t* and estimated the standard deviation across tasks (Fig. 3C). We summed all vertex-wise standard deviation values within the area and divided it by the number of vertices in this area to control for the effect of parcel size. We examined the correlation between intra- and inter-areal cross-task diversity by computing Spearman's correlation coefficient, controlling tSNR as a covariate, and controlling for the spatial autocorrelation using 1000 spin permutation tests.

### Reporting summary
Further information on research design is available in the Nature Portfolio Reporting Summary linked to this article.

### Data availability
The MRI data of the 7 T discovery dataset is openly available at the OSF platform (https://osf.io/mhq3f/)[58]. The Julich-Brain atlas is available at the EBRAINS platform (https://www.ebrains.eu/tools/human-brain-atlas). The MICA-MICs replication data is openly available at https://portal.conp.ca/dataset?id=projects/mica-mics [61]. Source data are pro-vided with this paper.

### Code availability
Gradient mapping analyzes was based on BrainSpace (https://brainspace.readthedocs.io/en/latest/)[129]. Code for MRI data pre-processing is available at https://github.com/MICA-MNI/micapipe [130]. The code for connectome gradients generation is available at https://github.com/MICA-MNI/BrainSpace. The Code for generating gradients used in this study, along with the main analysis is openly available on https://github.com/MICA-MNI/Wang_MultimodalGradient.

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

## Acknowledgements

Yezhou Wang, Dr. Alan Evans, Dr. Katrin Amunts, and Dr. Boris Bernhardt were supported by the Helmholtz International BigBrain Analytics and Learning Laboratory (HIBALL). Yezhou Wang was funded by the Fonds de recherche du Québec-Nature and technologies (FRQNT). This project/research/publication has received funding from the European Union's Horizon Europe Program under the Specific Grant Agreement No. 101147319 (EBRAINS 2.0 Project; KA). Dr. Nicole Eichert was supported by a Sir Henry Wellcome Postdoctoral Fellowship from the Wellcome Trust [222799/Z/21/Z]. Dr. Jessica Royer was supported by a fellowship from the Canadian Institute of Health Research (CIHR). Dr. Robert Leech was funded by the NIHR Maudsley Biomedical National Research Centre. Dr. Sofie Valk was funded by the Max Planck Institute. Dr. Boris Bernhardt furthermore acknowledges research support from the National Science and Engineering Research Council of Canada (NSERC Discovery-1304413), CIHR (FDN-154298, PJT-174995, PJT-191853), SickKids Foundation (NI17-039), Azrieli Center for Autism Research (ACAR-TACC), BrainCanada (Future-Leaders), and the Tier-2 Canada Research Chairs (CRC) program.

## Author contributions

Y.W. conceptualized the project, designed the methods, performed the analyzes, and drafted the manuscript. N.E. provided code and assistance for structural connectome estimate. C.P. performed analysis on histological dataset. R.R.-C., J.D., J.R., D.G.C., H.A., A.N., I.R.L., C.L.T., and D.A.R. helped to acquire and preprocess the imaging datasets. R. L. provided insights and interpretations of the statistical analysis. K.A., S.L.V., and J.S. were consulted with regards to the structural and functional imaging analyzes and helped to revise the manuscript. A.C.E. and K.A. provided detailed insights and interpretations of the cytoarchitectural analyzes. B.C.B. conceptualized the project, oversaw its execution, and edited the manuscript.

## Competing interests

The authors declare no competing interests.

## Ethical approval

We confirm that all relevant ethical regulations were adhered to in the design and conduct of this study.
