## [Transparent Peer Review file · Nature Communications]

Multimodal gradients unify local and global cortical organization

Corresponding Author: Mr Yezhou Wang

Version 0:

Reviewer comments:

Reviewer #1

(Remarks to the Author)

Thank you for inviting me to review this manuscript by Wang and colleagues, in which the authors conducted a robust, comprehensive comparison of a recent dataset characterising human brain structure at high resolution using post mortem data with a densely-sampled functional neuroimaging dataset collected on 10 individuals. The authors provide a series of thorough experiments that validate the relationship between measures calculated indirectly from human functional neuroimaging data and the high-resolution post mortem data. There are numerous checks and balances throughout the manuscript, including multiple successful replications and clear, sensible methodological choices that will make further extension and generalisation of these results possible.

Given the well-reasoned set-up, the clear exposition of the methods and the clarity of the results, I am happy to recommend publication in its current form.

(Remarks on code availability)

Reviewer #2

(Remarks to the Author)

In this work, Wang et al. use gradients to investigate the patterning of feature (dis)similarity and functional flexibility across the cortex. They derive cortical feature gradients from depth-wise quantitative T1 values, structural connectivity matrices, and functional connectivity profiles, and use these multi-modal gradients to define a main axis of cortical feature variability and inter-areal feature dissimilarity. They demonstrate that inter-areal dissimilarity decreases from idiosyncratic to paralimbic regions along a sensory-fugal cortical hierarchy and is lowest amongst brain regions that exhibit the most variable patterns of functional connectivity across task states. Furthermore, using the same multi-modal feature gradients, the authors study intra-area dissimilarity and suggest that it may be highest in heteromodal and unimodal cortices.

This work has numerous strengths. It approaches the study of regional feature (dis)similarity using three complementary imaging modalities. The main set of analyses utilizes high resolution, 7T MR data (N = 10) and key results are replicated in both an alternate 7T cohort (N = 12) and a 3T sample (N = 100), demonstrating robustness across field strengths and individuals. The paper is methodologically sound and the authors perform key sensitivity analyses.

However, I have questions concerning the interpretation of the main study findings, potential confounding factors, and how this work provides a conceptual advance over available literature.

The principal component of multi-modal gradient profiles (PCA axis)

- The authors apply a PCA to regional gradient profiles to derive an area-wise PCA axis (Fig. 1C) that captures variability in multi-modal gradients between different cortical areas. As the authors write in the text, the resulting PCA is "following a sensory-fugal axis". There is a wealth of excellent literature (spearheaded by this group and others) demonstrating that cortical feature variability is patterned along the cortical hierarchy along a sensory-fugal axis (Burt et al., 2018; García-Cabezas et al., 2019; Huntenburg et al., 2018; Margulies et al., 2016; Paquola et al., 2019; Sydnor et al., 2021). How does

the area-wise PCA axis (and the related inter-areal dissimilarity map) advance current knowledge on cortical organization and feature diversity?

- In Fig. 1C, the authors provide exemplar spider plots for cortical regions on the opposing ends and the middle of the area-wise PCA axis. However, it's not clear how these spider plots should be interpreted. What does it mean for example for the first region to have higher gradient z-scores for MPCG1, MPCG2, MPCG5, SCG2, but low scores for SCG1, FCG1, etc? What does it mean for the middle region to have a relatively more even distribution of gradient z-scores across the 15 gradients? Unpacking these spider plots and the gradient profiles that are characteristic of different portions of the area-wise PCA axis could help to provide valuable intuition into what this axis is capturing regarding regional feature profiles (and how it adds to the existing literature).

Inter-areal dissimilarity map and potential methodological confounds

- The authors derive an inter-areal dissimilarity map and show that it is highly reproducible across independent 7T samples and additionally can be replicated at 3T. It would be beneficial to demonstrate that the inter-areal dissimilarity map is not influenced by potential confounding scanner-related variables, especially those that vary between 7T and 3T, and differentiate somatomotor from medial paralimbic cortex. In particular, it would strengthen the work to demonstrate 1) that the inter-areal dissimilarity map is not related to B1+ transmit field inhomogeneity, which is prominent at 7T and influences quantitative T1 measures derived from MP2RAGE sequences (Haast et al., 2016), and 2) that the inter-areal dissimilarity map is not correlated with tSNR of the functional data, SNR of the diffusion data, or CNR of the T1 data.

Cortical patterning of intra-areal dissimilarity

- The authors use a stringent spatial permutation test procedure to identify cortical regions with significantly higher or lower inter-areal dissimilarity than the rest of the cortex (regions with values that are higher or lower than 97.5% of permuted values). Was this same procedure applied to test whether any region showed significantly higher or lower intra-areal dissimilarity than the rest of the cortex?

- It would be helpful to see individual region data points overlaid on the boxplot in Fig. 2D (and other boxplots) to understand whether higher intra-areal dissimilarity in heteromodal and unimodal regions as compared to paralimbic regions is driven by only a few regions. Relatedly, in the Discussion the authors write "we also observed elevated intra-areal dissimilarity in uni- and heteromodal association cortices, in line with their more integrative functional role and increased functional flexibility, compared to primary sensory and motor areas". Given that heteromodal and unimodal cortex did not show significantly higher intra-areal dissimilarity than idiosyncratic cortex, perhaps this interpretation should be more nuanced? Is intra-areal dissimilarity really a clear principle separating association cortex from paralimbic/idiosyncratic cortex, or are these differences accounted for by a small subset of areas with high dissimilarity as compared to the rest of the cortex?

- Is regional intra-areal dissimilarity influenced by the size of a cortical parcel (i.e., parcel surface area) or the average distance between vertices that make up the parcel?

Conflating cortical feature similarity and cortical integration

- A central premise of the current manuscript is that cortical regions that show more similar feature profiles to other cortical regions (low inter-areal dissimilarity) are also more globally integrative. For example, the Discussion states that "Notably, the pattern of cortical organization within the limbic system closely aligns with that of other cortical areas... This supports the integrative role of the limbic system". However, the theoretical foundations and empirical evidence linking feature similarity to global functional integration are not made clear in the manuscript. Isn't it possible, for example, that regions of the cortex that are remarkably integrative could show unique patterns of functional and structural connectivity and unique microstructural properties, which together enable their highly integrative nature and lead to high inter-areal dissimilarity? Relatedly, is it possible that brain regions with very low cross-task FC diversity show a highly integrative connectivity architecture that is stable across all tasks, and thus that high cross-task diversity may not always support "integrating specialized brain networks"? Perhaps the conclusions drawn in the manuscript linking feature (dis)similarity and cross-task functional diversity to global functional integration could be strengthened by directly computing a measure of segregated versus integrated processing from the functional connectivity data (e.g., the participation coefficient to measure diversity of connections (Bertolero et al., 2017) or global connectivity strength).

- The first paragraph of the manuscript states that "Global integration prominently manifests within higher-order systems, notably the transmodal association cortex, which engages in increasingly abstract and self-generated cognition. In contrast, local functional specialization is more frequent in sensory and motor regions that interact more closely with the here and now." Can the authors elaborate on how the present findings extend these statements?

(Remarks on code availability)

The authors provide links to software packages used for preprocessing (micapipe) and derivation of multi-modal gradients (brainspace). However, no code is provided demonstrating how this software was applied to generate the data used in the present manuscript; this somewhat limits reproducibility. The code provided for the "main analysis" is sufficient. The release of data used in the manuscript is valuable.

Version 1:

Reviewer comments:

Reviewer #2

(Remarks to the Author)

The authors satisfactorily addressed all of my original comments in the revision. This work is a nice contribution to the field.

(Remarks on code availability)

The authors improved the code availability by including a demo script to build vertex-wise gradients.

Response to the Reviewers (NCOMMS-24-55409)

We would like to thank the Editors for the opportunity to submit a revised manuscript, as well as the Editors and Reviewers for their thoughtful evaluations of our paper. We are grateful for their constructive comments and found the suggestions very helpful to improve the quality of our paper. We addressed all suggestions in a point-by-point fashion and highlighted the corresponding changes in the manuscript in red.

Reviewer #1:

R1#1. Thank you for inviting me to review this manuscript by Wang and colleagues, in which the authors conducted a robust, comprehensive comparison of a recent dataset characterising human brain structure at high resolution using post mortem data with a densely-sampled functional neuroimaging dataset collected on 10 individuals. The authors provide a series of thorough experiments that validate the relationship between measures calculated indirectly from human functional neuroimaging data and the high-resolution post mortem data. There are numerous checks and balances throughout the manuscript, including multiple successful replications and clear, sensible methodological choices that will make further extension and generalisation of these results possible.

Given the well-reasoned set-up, the clear exposition of the methods and the clarity of the results, I am happy to recommend publication in its current form.

We thank Reviewer 1 for the very positive evaluation of our paper.

Reviewer #2:

R2#1. In this work, Wang et al. use gradients to investigate the patterning of feature (dis)similarity and functional flexibility across the cortex. They derive cortical feature gradients from depth-wise quantitative T1 values, structural connectivity matrices, and functional connectivity profiles, and use these multi-modal gradients to define a main axis of cortical feature variability and inter-areal feature dissimilarity. They demonstrate that inter-areal dissimilarity decreases from idiosyncratic to paralimbic regions along a sensory-fugal cortical hierarchy and is lowest amongst brain regions that exhibit the most variable patterns of functional connectivity across task states. Furthermore, using the same multi-modal feature gradients, the authors study intra-area dissimilarity and suggest that it may be highest in heteromodal and unimodal cortices.

This work has numerous strengths. It approaches the study of regional feature (dis)similarity using three complementary imaging modalities. The main set of analyses utilizes high resolution, 7T MR data (N = 10) and key results are replicated in both an alternate 7T cohort (N = 12) and a 3T sample (N = 100), demonstrating robustness across field strengths and individuals. The paper is methodologically sound and the authors perform key sensitivity analyses.

We thank Reviewer 2 for the positive evaluation of our paper.

R2#2. However, I have questions concerning the interpretation of the main study findings, potential confounding factors, and how this work provides a conceptual advance over available literature. We thank Reviewer 2 for the thought- and helpful comments, which we addressed point-by-point below.

R2#3. The principal component of multi-modal gradient profiles (PCA axis)

- The authors apply a PCA to regional gradient profiles to derive an area-wise PCA axis (Fig. 1C) that captures variability in multi-modal gradients between different cortical areas. As the authors write in the text, the resulting PCA is “following a sensory-fugal axis”. There is a wealth of excellent literature (spearheaded by this group and others) demonstrating that cortical feature variability is patterned along the cortical hierarchy along a sensory-fugal axis (Burt et al., 2018; García-Cabezas et al., 2019; Huntenburg et

al., 2018; Margulies et al., 2016; Paquola et al., 2019; Sydnor et al., 2021). How does the area-wise PCA axis (and the related inter-areal dissimilarity map) advance current knowledge on cortical organization and feature diversity?

We thank Reviewer 2 for the insightful comments. By estimating the area-wise PCA axes, our study provides novel evidence that the sensory-fugal axis is not confined to a single modality (*e.g.*, functional or structural MRI) but that it extends to brain networks shaped by multimodal inputs. This finding underscores how this hierarchy jointly governs the organizational principles of microstructure, structure, and function. To better explain these results, we updated the *Results* (P.6):

“This approach integrates salient features of its constituents (i.e., the individual MPC, SC, and FC gradients) in a synoptic manner, suggesting an overarching principle of cortical organization across multiple modalities.”

And the *Discussion* (P.11):

“The first principal component of our estimated multimodal gradient profiles demonstrated that sensory-functional axes jointly guide hierarchical patterns of cortical microstructure, structure, and function. Our work offers one of the first descriptions of a converging overarching principle, extending prior work that has focussed on specific modalities in isolation^{19,20,70}. These findings suggested a convergence of organizational principles across different scales and at the level of structure and function. Notably, and in addition to harnessing a multimodal imaging approach, we leveraged a recently disseminated atlas derived from post-mortem histological data to finely partition the brain into distinct areas based on cytoarchitecture, a presumed gold standard for cortical mapping¹⁶. Our work, thereby, integrated dimensional gradient and area-based descriptions of macroscale cortical spatial patterns, providing a unified framework for understanding the cortical layout. This framework could identify key principles of between- and within-parcel heterogeneity, distinguish different zones across the cortical hierarchy, and reveal associations between these heterogeneity patterns and cross-task diversity.”

R2#4. - In Fig. 1C, the authors provide exemplar spider plots for cortical regions on the opposing ends and the middle of the area-wise PCA axis. However, it's not clear how these spider plots should be interpreted. What does it mean for example for the first region to have higher gradient z-scores for MPCG1, MPCG2, MPCG5, SCG2, but low scores for SCG1, FCG1, etc? What does it mean for the middle region to have a relatively more even distribution of gradient z-scores across the 15 gradients? Unpacking these spider plots and the gradient profiles that are characteristic of different portions of the area-wise PCA axis could help to provide valuable intuition into what this axis is capturing regarding regional feature profiles (and how it adds to the existing literature).

We thank Reviewer 2 for the comments. We noticed that the labels and sequence of the spider plots in Fig. 1C were not updated, which may have caused some confusion. We revised the figure to ensure accurate correspondence between the spider plots and the sorted gradient profiles.

C. Area-wise gradient profiling

Additionally, we interpreted these spider plots and updated the *Results* (P.6):

“The reordered gradient profiles located in the middle and at the two ends of the main axis were examined and showed diverse patterns (Fig. 1C). Specifically, we observed that the bottom region of the PCA axis, corresponding to sensorimotor areas, exhibited lower gradient z-scores for MPCG1 and FCG1 and higher scores for SCG1 and SCG2. This pattern suggests that the sensorimotor network represents one end of the hierarchy across all modalities. While the direction of structural gradients differed from functional and microstructural gradients, the first gradient across modalities consistently captured the most variance. At the opposite end of the PCA axis, regions in the inferior frontal sulcus showed the reverse pattern, representing the other extreme of the hierarchy. In contrast, the middle region displayed a relatively uniform z-score distribution, suggesting its role in linking higher-order and lower-order regions for multimodal information processing.”

R2#5. Inter-areal dissimilarity map and potential methodological confounds

- The authors derive an inter-areal dissimilarity map and show that it is highly reproducible across independent 7T samples and additionally can be replicated at 3T. It would be beneficial to demonstrate that the inter-areal dissimilarity map is not influenced by potential confounding scanner-related variables, especially those that vary between 7T and 3T, and differentiate somatomotor from medial paralimbic cortex. In particular, it would strengthen the work to demonstrate 1) that the inter-areal dissimilarity map is not related to B1+ transmit field inhomogeneity, which is prominent at 7T and influences quantitative T1 measures derived from MP2RAGE sequences (Haast et al., 2016), and 2) that the inter-areal dissimilarity map is not correlated with tSNR of the functional data, SNR of the diffusion data, or CNR of the T1 data. We thank Reviewer 2 for the insightful comments. In response, we now estimated (i) B1+ transmit inhomogeneity using dedicated acquisitions, (ii) the SNR map from the used diffusion scans, (iii) the CNR map from the used T1 maps, and (iv) the tSNR map from the functional data. These maps were sampled onto the cortical surface, and spatially correlated with the inter-areal dissimilarity map. Results indicated that inter-areal dissimilarity was correlated with tSNR ($r=0.66$, $p<0.001$), but not with B1+ ($r=-0.05$, $p=0.45$), CNR ($r=0.12$, $p=0.07$), or SNR ($r=-0.12$, $p=0.08$) maps. Notably, reassessing the association between inter-areal dissimilarity and cross-task diversity remained robust even when controlling for tSNR

($r=-0.46$, $p<0.001$), $B1+$ ($r=-0.71$, $p<0.001$), CNR ($r=-0.70$, $p<0.001$), and SNR ($r=-0.71$, $p<0.001$), respectively. We incorporated these control analyses into the revised *Results* (P.7; P.8):

“Although inter-areal dissimilarity was found to correlate with temporal signal-to-noise ratio (tSNR) map ($r=0.66$, $p<0.001$; Supplementary Table 1), no significant correlations were observed with other technical imaging metrics, including the $B1+$ field map ($r=-0.05$, $p=0.45$), the contrast-to-noise ratio (CNR) map from T1 scans ($r=0.12$, $p=0.07$), or the SNR map from diffusion scans ($r=-0.11$, $p=0.08$)..”

“Notably, we identified a marked correlation between cross-task diversity and inter-areal dissimilarity ($\rho=-0.71$, $p_{spin}<0.001$; Fig. 3B). To control for potential influences from image quality, we performed partial correlation analyses. The results remained consistent when controlling for tSNR ($\rho=-0.46$, $p_{spin}<0.001$), $B1+$ field map ($r=-0.71$, $p<0.001$), CNR from the T1 data ($r=-0.70$, $p<0.001$), and SNR from the diffusion data ($r=-0.71$, $p<0.001$).”

Supplementary Table 1

	Correlations with inter-areal dissimilarity		Partial correlations between inter-areal dissimilarity and cross-task diversity	
	r	p	r	P _{spin}
B1+ field	-0.05	0.45	-0.71	<0.001
T1 CNR	0.12	0.07	-0.70	<0.001
DWI SNR	-0.11	0.08	-0.71	<0.001
fMRI tSNR	0.66	<0.001	-0.46	<0.001

and *Methods* (P.17)

“The MP2RAGE scans of each subject were reoriented using FSL¹⁰³, ... The $B1+$ field map and the CNR of the T1 map were calculated for each participant to assess reliability.”

“Regarding the DWI data, pre-processing was carried out using MRtrix¹⁰⁷ in the native DWI space. ... We computed the mean and standard deviation maps of the b_0 images. For each participant, voxel-wise SNR maps were generated by dividing the mean map by the standard deviation map. These SNR maps were subsequently projected onto the fsLR-5k surface.”

“For the rs-fMRI scans, pre-processing steps were conducted using AFNI¹¹² and FSL¹⁰³ tools. ... The tSNR maps were estimated for each participant.”

and updated the *Discussion* (P.11).

“Reliability analyses indicated that this inter-areal pattern was not influenced by potential methodological biases, including the $B1+$ field map, SNR of diffusion data, or CNR of T1 data. While tSNR was correlated with inter-areal dissimilarity, the correlation between inter-areal dissimilarity and cross-task diversity remained significant even after controlling for tSNR.”

R2#6. Cortical patterning of intra-areal dissimilarity

- The authors use a stringent spatial permutation test procedure to identify cortical regions with significantly higher or lower inter-areal dissimilarity than the rest of the cortex (regions with values that are higher or lower than 97.5% of permuted values). Was this same procedure applied to test whether any region showed significantly higher or lower intra-areal dissimilarity than the rest of the cortex?

Following the Reviewer’s suggestion, we also performed spin tests when assessing intra-areal dissimilarity. Our findings revealed that the medial supplementary motor area exhibited the highest dissimilarity, whereas the medial orbitofrontal cortex showed the lowest. Please see the updated *Results* (P.7):

“The medial supplementary motor areas exhibited highest intra-areal dissimilarity ($p_{spin} < 0.05$, spatial permutation tests, FDR correction), while the medial orbitofrontal cortex showed the lowest values under the same statistical threshold.”

We also updated Fig.2 C to highlight the regions with the highest intra-areal dissimilarity.

C. Intra-areal dissimilarity patterns

As well as the legend of Fig.2 C:

“Regions with significantly higher intra-areal dissimilarity compared to other areas after applying FDR correction were highlighted using black boundaries.”

R2#7. - It would be helpful to see individual region data points overlaid on the boxplot in Fig. 2D (and other boxplots) to understand whether higher intra-areal dissimilarity in heteromodal and unimodal regions as compared to paralimbic regions is driven by only a few regions. Relatedly, in the Discussion the authors write “we also observed elevated intra-areal dissimilarity in uni- and heteromodal association cortices, in line with their more integrative functional role and increased functional flexibility, compared to primary sensory and motor areas”. Given that heteromodal and unimodal cortex did not show significantly higher intra-areal dissimilarity than idiosyncratic cortex, perhaps this interpretation should be more nuanced? Is intra-areal dissimilarity really a clear principle separating association cortex from paralimbic/idiosyncratic cortex, or are these differences accounted for by a small subset of areas with high dissimilarity as compared to the rest of the cortex?

We thank Reviewer 2 for the insightful comments. As mentioned in our manuscript, “we calculated the cosine distance between the gradient profile of each cortical vertex and the mean gradient profile of the area to which it belongs, as a measure of intra-areal dissimilarity (Fig. 2C).” Accordingly, we added the vertex-wise data points to the updated Fig. D and updated the figure legend:

D. Intra-areal dissimilarity in cortical hierarchies and associations with histological gradient

“D The distribution of vertex-wise intra-areal dissimilarity in four cortical hierarchies is shown in the left panel.”

We agree with the Reviewer’s impression that there was no clear separation in terms of dissimilarity across different levels. Vertex-wise measures fell within the same range, and their distributions showed substantial overlap. However, we observed a trend toward higher dissimilarity in unimodal+heteromodal regions, as well as a greater proportion of vertices with elevated values. We updated the *Results* (P.8):

*“In fact, there was a trend indicating that intra-areal dissimilarity was higher in heteromodal and unimodal association systems compared to idiotypic and paralimbic regions, though this difference was only marginally significant ($p_{spin}=0.042$, $p_{spin}=0.045$, FDR correction; **Fig. 2D**).”*

We added further nuance to the *Discussion*, as suggested (P.13):

“Notably, although there was no clear difference in intra-areal dissimilarity between networks, we observed a trend towards increased dissimilarity within the unimodal and heteromodal cortices. We speculate that this may be related to their more integrative functional role and increased functional flexibility compared to primary sensory and motor areas, which deserves further verification in future work.”

We also updated the *Fig. B*:

B. Inter-areal dissimilarity in cortical hierarchies and associations with histological gradient

R2#8. - Is regional intra-areal dissimilarity influenced by the size of a cortical parcel (i.e., parcel surface area) or the average distance between vertices that make up the parcel? We calculated the surface area of each vertex and used this to estimate the overall size of each region in the Julich-Brain atlas. This measure was overall correlated to intra-areal dissimilarity ($\rho=0.44$, $p_{spin}<0.001$).

Similarly, average geodesic distance between vertices within each parcel was also correlated with intra-areal dissimilarity ($\rho=0.56$, $p_{spin}<0.001$).

We updated the *Results* (P.8) to report this association with parcel size:

“Additionally, we found that intra-areal dissimilarity was positively correlated with parcel size ($\rho=0.44$, $p_{spin}<0.001$).”

For our estimate of intra-areal dissimilarity, we accounted for the effect of parcel size by calculating the mean vertex-wise intra-areal dissimilarity within each region. While we acknowledge that parcel size may still exert some influence, we believe this association is both reasonable and expected. The difference in intra-areal dissimilarity between idiosyncratic and unimodal/heteromodal networks remained marginally significant when controlling for parcel size using ANCOVA ($t=2.07$, $p=0.039$; $t=1.93$, $p=0.054$). Moreover, we observed significant difference in intra-areal dissimilarity between idiosyncratic and paralimbic networks when controlling for the average geodesic distance ($t=-2.20$, $p=0.029$).

We thus updated the *Discussion* (P.) to emphasize the potential influence of parcel size:

“Meanwhile, the potential effects of region size should also be considered, as larger regions are more likely to exhibit higher intra-region dissimilarity.”

As noted in the *Results*, we controlled for parcel size when assessing the relationship between intra-areal dissimilarity and functional diversity across task states: *“Here, we defined intra-areal cross-task diversity as the standard deviation between vertex-wise FC across different tasks, with the effect of parcel size controlled.”*

R2#9. Conflating cortical feature similarity and cortical integration

- A central premise of the current manuscript is that cortical regions that show more similar feature profiles to other cortical regions (low inter-areal dissimilarity) are also more globally integrative. For example, the Discussion states that “Notably, the pattern of cortical organization within the limbic system closely aligns with that of other cortical areas... This supports the integrative role of the limbic system”. However, the theoretical foundations and empirical evidence linking feature similarity to global functional integration are not made clear in the manuscript. Isn't it possible, for example, that regions of the cortex that are remarkably integrative could show unique patterns of functional and structural connectivity and unique microstructural properties, which together enable their highly integrative nature and lead to high inter-areal dissimilarity? Relatedly, is it possible that brain regions with very low cross-task FC diversity show a highly integrative connectivity architecture that is stable across all tasks, and thus that high cross-task diversity may not always support “integrating specialized brain networks”? Perhaps the conclusions drawn in the manuscript linking feature (dis)similarity and cross-task functional diversity to global functional integration could be strengthened by directly computing a measure of segregated versus integrated processing from the

functional connectivity data (e.g., the participation coefficient to measure diversity of connections (Bertolero et al., 2017) or global connectivity strength).

We thank Reviewer 2 for these insightful comments. As suggested, we examined the relationships between inter-areal dissimilarity and participation coefficient and global functional connectivity strength based on group-level task FC. Results suggest positive, albeit non-significant correlations (PC: $\rho=0.13$, $p_{spin}>0.05$; global FC strength: $\rho=0.32$, $p_{spin}>0.05$). Similar findings were observed when calculating these metrics based on group-level resting-state FC (PC: $\rho=0.13$, $p_{spin}>0.05$; global FC strength: $\rho=0.36$, $p_{spin}>0.05$). We included these findings in the revised *Results* (P.7):

“To investigate functional integration patterns, we estimated the participation coefficient (PC)⁶⁰ and global FC strength. Positive, but non-significant, correlations were observed between these metrics and inter-areal dissimilarity (PC: $\rho=0.13$, $p_{spin}>0.05$; global FC strength: $\rho=0.32$, $p_{spin}>0.05$). Similar findings were observed when calculating these metrics based on group-level resting-state FC (PC: $\rho=0.13$, $p_{spin}>0.05$; global FC strength: $\rho=0.36$, $p_{spin}>0.05$).”

We also updated the *Methods* (P.21):

“To investigate the relationship between inter-areal dissimilarity and functional integration patterns, we calculated the PC⁶⁰ using four cortical hierarchical levels⁶¹ as communities. Additionally, we estimated global FC strength by averaging the FC values for each region.”

As suggested, we provided a more nuanced discussion on the associations between cortical heterogeneity, global functional integration, and functional diversity. We updated the *Discussion* (P.12):

“For instance, the paralimbic system with more long-distance connections was reported to have higher microstructural similarity⁵⁷. Similarly, another study combining structural and functional connections revealed that less diverse unimodal regions show a preference for local communication, while more diverse multimodal regions engage in more global communication patterns⁷². Indeed, the degree of laminar differentiation also varies gradually across the cortex, reflecting the specialization of underlying cortical microcircuits^{73,74}. It is highest in primary areas, such as the visual and somatosensory cortices, and then decreases in unimodal and heteromodal regions, to reach its lowest level in agranular areas^{7,75}. Projection patterns similarly follow this gradient⁷⁶⁻⁷⁸. Together, these organizational patterns, spanning from the microscale to the macroscale may support the integrative role of the limbic system, enabling it to participate in various cognitive processes⁷⁹. However, it is important to note that no significant associations were observed between PC⁶⁰ and either inter-areal dissimilarity or cross-task functional diversity, suggesting a more complex relationship between cortical heterogeneity, functional diversity, and different indices of macroscale functional integration.”

R2#10. - The first paragraph of the manuscript states that “Global integration prominently manifests within higher-order systems, notably the transmodal association cortex, which engages in increasingly abstract and self-generated cognition. In contrast, local functional specialization is more frequent in sensory and motor regions that interact more closely with the here and now.” Can the authors elaborate on how the present findings extend these statements?

We elaborate on how our findings extend these statements and updated the *Discussion* (P.10):

“We noted higher inter-areal dissimilarity in sensorimotor cortices and lower inter-areal dissimilarity in the transmodal system, indicating the distinctiveness of the primary sensory cortex as a functionally specialized system. Additionally, functional connectivity in the primary cortex exhibited less variability across tasks, suggesting that information processing from different tasks may converge at early stages. Significant associations were identified between cross-task functional diversity and inter-areal gradient profile dissimilarity, indicating a link between global cortical motifs and functional flexibility across different task contexts.”

Reviewer #2 (Remarks on code availability):

R2#11. The authors provide links to software packages used for preprocessing (micapipe) and derivation of multi-modal gradients (brainspace). However, no code is provided demonstrating how this software was applied to generate the data used in the present manuscript; this somewhat limits reproducibility. The code provided for the "main analysis" is sufficient. The release of data used in the manuscript is valuable.

The vertex-wise microstructural, structural and functional connectome data used to generate the gradients studied in this work are available on the OSF platform (<https://osf.io/mhq3f/>). Additionally, we have provided demo code for generating cortical gradients used from these vertex-wise connectome data using Brainspace, on our Github repository (https://github.com/MICA-MNI/micaopen/tree/master/gradient_profiles). The resulting vertex-wise gradients are identical to those used in this manuscript. We updated the *Code Availability* (P.22):

“The code for generating gradients used in this study, along with the main analysis is openly available on https://github.com/MICA-MNI/Wang_MultimodalGradient.”